# A Machine Learning-Driven Electrophysiological Platform for Real-Time Tumor-Neural Interaction Analysis and Modulation

Ting Xu[1,2,8], Xinyue Zhang[2,8], Youheng Jiang[3,8], Kai Sheng[2,8], Jie Li[1,2], Jinliang Ren[1,2], Jiahao He[1,2], Chaofeng Liang[4], Zhenhua Yu[5], Huawei Jin[5], Bowen Zhuang[6], Lujing Li[7], Ningning Li [3] & Bingzhe Xu [1,2] ✉

Neural-tumor electrophysiology—marked by pathological membrane potentials and ion channel dysregulation—emerges as actionable targets to curb tumor aggression. Yet, how neural-driven bioelectrical crosstalk dynamically regulates tumors within functional circuits remains elusive, demanding tools for real-time interaction decoding. Here, we present a machine learning-driven electrophysiological platform that integrates custom microfluidics with real-time decoding of complex neural-tumor signal dynamics. Our findings show that glioma cells selectively hijack specific subsets of neural signals, reshaping waveform properties and synchronizing their firing events with neural activity. This dynamic interaction plays a critical role in boosting glioma invasiveness, as tumor cells harness neural activity to promote their progression. Notably, targeted stimulation of glioma cells with these hijacked signal patterns—without direct neural involvement—is sufficient to induce hyper-invasive behavior, emphasizing the role of these electrical cues as drivers of tumor aggression.

Understanding tumor-neural interactions is critical to uncovering the mechanisms driving glioma progression and invasiveness. Recent studies reveal that electrophysiological properties of tumor cells—such as membrane potential and ion channel activity—are central regulators of proliferation, migration, and invasion[1–8] These properties, integral to the tumor microenvironment[9], influence cancer behavior by modulating ion permeability and signal transduction, leading to abnormal membrane polarization[3,10,11] Ion channels thus represent promising therapeutic targets[12–14], as they control proliferation via membrane potential and ionic homeostasis at cell cycle checkpoints (e.g., voltage-gated sodium channels maintaining glioblastoma stemness in G0 phase)[15,16], and calcium-activated potassium channels (BK) correlate with glioma grade and growth[17]. For migration and invasion, coordinated $K^+/Cl^-$ efflux through BK channels enables

[1]School of Biomedical Engineering, Sun Yat-sen University, No. 135, Xingang Xi Road, Guangzhou 510275, P.R. China. [2]School of Biomedical Engineering, Shenzhen Campus of Sun Yat-sen University, No.66, Gongchang Road, Guangming District, Shenzhen 518107, P.R. China. [3]Tomas Lindahl Nobel Laureate Laboratory, The Seventh Affiliated Hospital of Sun Yat-Sen University, 628 Zhenyuan Road, Shenzhen 518107, P.R. China. [4]Department of Neurosurgery, The Third Affiliated Hospital of Sun Yat-sen University, No.600 Tianhe Road, Guangzhou 510080, P.R. China. [5]Department of Neurosurgery, The First Affiliated Hospital of Sun Yat-sen University, No.58 Zhongshan Er Road, Guangzhou 510080, P.R. China. [6]Department of Ultrasound, The First Affiliated Hospital of Sun Yat-sen University, No.58 Zhongshan Er Road, Guangzhou 510080, P.R. China. [7]Department of Ultrasound, The Seventh Affiliated Hospital, Sun Yat-sen University, 628 Zhenyuan Road, Shenzhen 518107, China. [8]These authors contributed equally: Ting Xu, Xinyue Zhang, Youheng Jiang, Kai Sheng. ✉e-mail: xubzh5@mail.sysu.edu.cn

volume regulation for tissue navigation[18,19], while calcium transients drive collective invasion[20]. Glioblastomas exhibit pathological ion channel reprogramming, including Kir4.1 loss and BK upregulation[21], with ~90% of ion channel mutations linked to aggressive phenotypes through disrupted differentiation and enhanced plasticity[22]. Mutations in ion channels and pumps are common in GBM[23], promoting high proliferation, migration, and invasiveness[24].

One particularly compelling aspect of tumor electrophysiology is the bidirectional interaction between neurons and glioma cells[5,25]. Neuronal activity promotes glioma progression via synaptic integration and neurotrophic factor release, as glioma cells can structurally and electrically incorporate into neural circuits[26]. Neurons modulate glioma malignancy through neural excitation, paracrine signaling, and direct synaptic connections[27–30]. Paracrine factors such as neuroligin-3 and BDNF[31], along with neuron-to-glioma AMPA receptor–mediated synapses[32], induce excitatory postsynaptic currents (EPSCs) that depolarize glioma membranes and drive proliferation[33,34]. Functional glutamatergic "neurogliomal synapses" identified by Venkataramani et al[32]. demonstrate that presynaptic glutamate activates AMPA receptors on glioma cells, triggering EPSCs, depolarization, and calcium influx[35], thereby enhancing glioma stem cell (GSC) proliferation and invasion. Additionally, glioma networks generate rhythmic electrical waves propagated through tumor microtubes, reinforcing malignancy[3]. These neuron–glioma interactions remodel both cellular and network dynamics[36–39], amplifying glioma excitability and growth while increasing neuronal hyperexcitability through glutamate release and synaptogenic signaling, and reducing inhibitory interneurons[35,40–42] Therapeutically, targeting these pathways shows promise: AMPA receptor antagonists such as perampanel block neuron-induced EPSCs and suppress tumor growth[43]; anti-inflammatory agents like meloxicam reduce invasiveness by disrupting gap junction–mediated $Ca^{2+}$ signaling; and gabapentin inhibits glioma-driven synaptogenesis by blocking TSP1 signaling[44].

Despite these advances[45–50] current research tools lack the resolution and versatility to systematically study complex interactions between neurons and tumors. In this work, we present a machine learning–driven electrophysiological platform that enables real-time analysis and modulation of tumor–neural interactions. Integrating microfluidic precision with computational modeling, this system decodes and manipulates complex electrophysiological signals to reveal high-resolution dynamics of glioma–neuron communication. Our approach identifies specific neural circuits hijacked by glioma cells to promote hyper-invasive behavior and uncovers distinct electrophysiological signatures associated with this invasive state. Moreover, we demonstrate that reproducing these signatures in vitro recapitulates tumor phenotypes, supporting the hypothesis that bidirectional neuron–glioma signaling exerts a stronger influence on invasiveness than neural activity alone.

## Results

### Platform design and characterization

Here, we present a machine learning-driven electrophysiological platform that integrates custom microfluidics with real-time decoding of complex neural-tumor signal dynamics (Fig. S1). The platform architecture is composed of five distinct layers with specific functionalities: (1) Signal Acquisition Hardware Layer, featuring microelectrode arrays (MEAs) for capturing neural and tumor electrophysiological activities, paired with a stimulation device for precise electrical modulation; (2) Data Collection and Processing, which filters, denoises, and normalizes raw signals (processing 127,060 signals with spectral analysis and waveform characterization) to extract spatiotemporal tumor-neural correlations; (3) Machine Learning (LSTM Architecture), employing dual LSTM layers (128 units each) with a 0.3 dropout rate to decode local/global temporal dependencies, enabling pattern recognition via signal prediction

(ReLU, MSE) and classification (Softmax, CE); (4) Regulation and Intervention, identifying neural signal hijacking events and implementing real-time feedback control; (5) Visualization and Monitoring, providing dynamic tracking of tumor invasion and neural activity through integrated imaging and fluorescence staining. By synergizing microfluidics, MEAs, real-time analytics, and closed-loop intervention, this platform offers all-in-one solution for dissecting tumor-nerve crosstalk.

The microfluidic MEA component (Fig. 1a) integrates several critical components for precise tumor-neural interaction analysis. First, dual-channel perfusion inlets and outlets enable dynamic nutrient delivery and waste removal, mimicking physiological microenvironments while maintaining long-term co-culture stability. Then a high-density MEA layout with spatially optimized contact points ensures real-time electrophysiological recording from tumor-neural interface regions (Fig. 1a), capturing local field potentials (LFPs) and single-cell spiking activities. Tumor and neural systems are initially confined to separated regions (Fig. 1b), with micro-channels allowing real-time monitoring tumor invasion into neural networks over 7 days. The electrophysiological recordings obtained during this period showed distinct patterns that corresponded to different stages of the interaction between the tumor and nerve cells. Leveraging deep learning technology, the platform achieves feature-specific weighting for signal discrimination, and analyzes spatiotemporal correlations between tumor invasion and neural activity through LSTM-based dynamic interaction mapping (Fig. 1c).

A customized detection platform (Fig. S2), comprising a collection module, microscope port, signal amplification system, processing platform, and shielding enclosure, was designed to be compatible with sensor and imaging technologies (Fig. 2a). This integration enables the combination of neural imaging and fluorescence staining techniques, facilitating the acquisition of high-dimensional data on tumor and neural activities. The staggered electrode layout (Fig. 2b, c) allows real-time discrimination between newly invading tumor signals (detected at right electrodes with <10 ms latency) and embedded hybrid signals (recorded from central electrodes with time-delayed correlation to neural activity). By assigning unique electrode clusters to tumor-front regions, the system generates spatially encoded signal fingerprints, enabling machine learning algorithms to classify invasion stages based on signal propagation patterns (e.g., radial spread vs. neural co-option). As demonstrated in Fig. 2d, the platform's signal processing pipeline leverages these spatiotemporal tags to dynamically separate signals into different categories (signal 1, 2, 3..., indicating early invasion signals, mixed region signals, and established invasion signals). Figure 2e displays the fabricated device picture. This microfluidic-electrode array chip connects to external circuits via a specialized vertical quick-clamp-based connector composed of a base, circuit platform, and quick clamp (Fig. 2f, g). The acquisition board features four pre-stage probe interfaces, each capable of connecting two RHD-2000 pre-stage probes, thus supporting up to eight 64-channel pre-stage probes and acquiring signals from 512 channels simultaneously. Driven by an FPGA module, the board synchronizes input data from all pre-stage probes, serializes the parallel inputs, and transmits the data to the upper computer.

The electrode array demonstrated consistent resistance values across both manufacturing and assembly (Fig. 2h), ensuring reliable and uniform signal capture. Gaussian-distributed baseline noise (pH 7.2–7.4) confirmed predominantly internal system-derived interference, validating a high-sensitivity electrophysiological platform optimized for intracranial microenvironment simulation. Baseline noise (4.91 to −4.65 µV, Fig. S3) remained within neural signal detection thresholds, demonstrating a low-noise design essential for precise signal acquisition in microfluidic/co-culture systems where neural amplitudes are inherently small. Recorded neural electrical signals and processed data (Fig. 2i, j) demonstrate the system's effectiveness. We

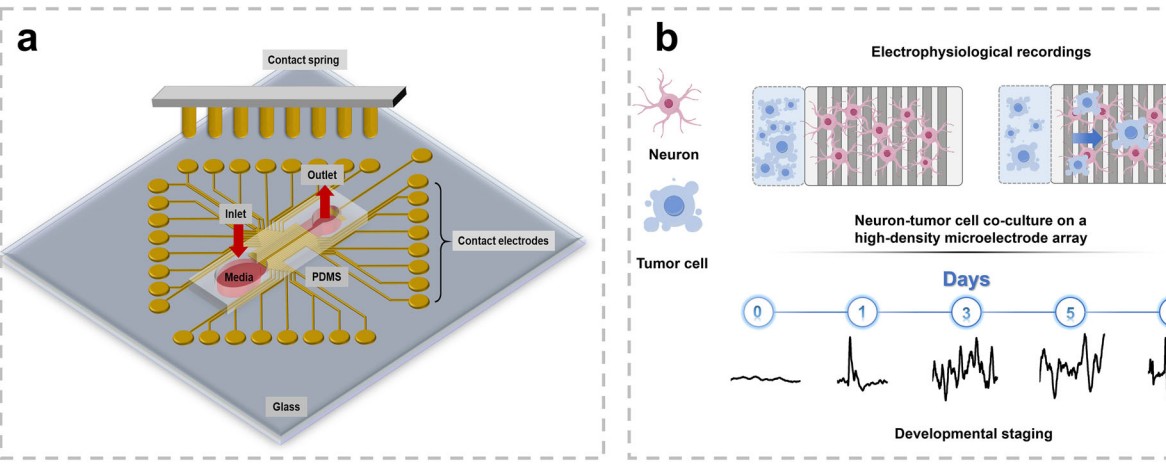

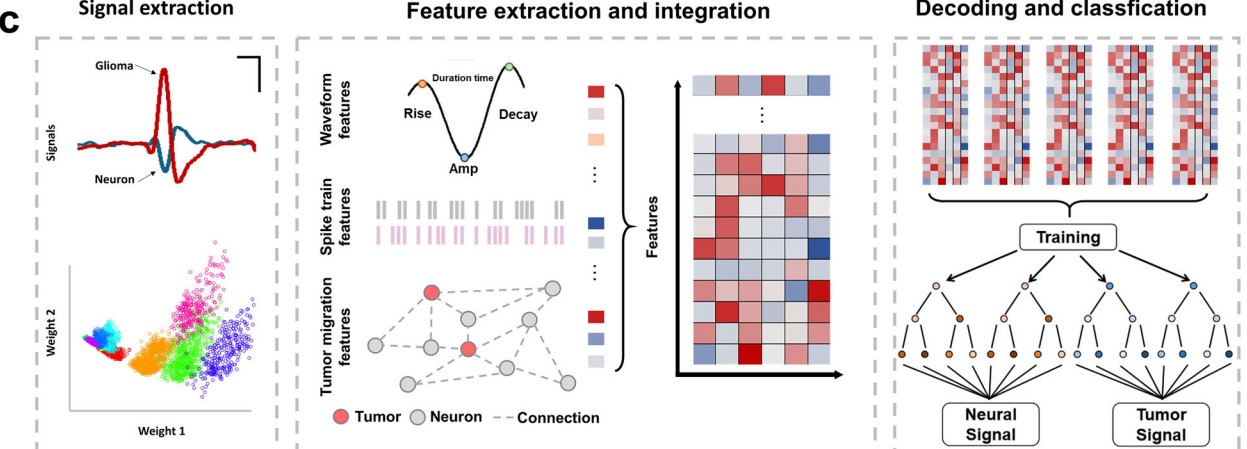

**Fig. 1 | The design of microfluidic MEA component and machine learning classification. a** Schematic design of the microfluidic MEA component. **b** Illustration of electrophysiological recordings, including the neuron-tumor cell co-culture on a high-density microelectrode array and developmental staging over days. **c** The process of signal extraction, feature extraction and integration, and decoding and classification in the tumor-neuron hybrid system.

implemented a modified threshold filter to mitigate signal peaks that could distort noise distribution estimates (Fig. 2k), achieving a 17% reduction in mean absolute error (MAE) post-filtering. The filtering criteria were as follows:

$$threshold = \text{AVG} + factor \cdot \text{STD} \tag{1}$$

$$\text{AVG}' = \frac{1}{n}\sum_{i}^{n}x_i' \tag{2}$$

$$\text{STD}' = \sqrt{\frac{\sum_{i}^{n}(x_i' - \text{AVG}')^2}{n}} \tag{3}$$

$$modified threshold = \text{AVG}' + factor \cdot \text{STD}' \tag{4}$$

$$x_i' \in signal \text{ and } |x_i'| < threshold$$

In summary, these results validate the platform's capability to capture fine-scale electrophysiological events during glioma invasion within a neural microenvironment. The minimized baseline noise level enhances data precision, essential for accurately detecting and interpreting dynamic signal changes in co-culture conditions.

## Hyper-invasive behavior in brain tumors driven by specific neural networks

We developed a pneumatic-based microfluidic system (Fig. 3a) designed to reliably initiate glioma cell invasion in a controlled neural environment. The system comprises a main chamber, microchannel, air pump interface, and pump. The open main chamber allows for direct loading of cells and culture medium, while the microchannel, connects the main chamber to the air pump interface, which controls air pressure within the microchannel to regulate the initial positioning of cells. The fluid control unit precisely managed liquid flow within the microchannel, improving the reliability and reproducibility of the experimental setup (Fig. 3a, MOVIE S1). To track cell locations effectively, the glioma cells were transfected with the GFP expression gene, enabling live cells to emit green fluorescence under 488 nm excitation (Fig. 3b). This setup allows for precise tracking of cell movements and interactions with the environment, thereby facilitating a detailed analysis of electrophysiological changes during tumor invasion. In our 24-h continuous observation of glioma cells within the microchannel, we observed obvious cellular movement (Fig. 3c), demonstrating the system's effectiveness in recording glioma cell migration.

Distinct populations of green-fluorescent glioma cells and neuron-marker-positive neural cells within the microfluidic channel (Fig. 3d) were arranged to facilitate direct interactions, which is critical for simulating the in vivo environment where glioma cells engage with neural networks. Over a 7-day observation period of temporal changes in glioma cell distribution across electrodes, we observed hyper-

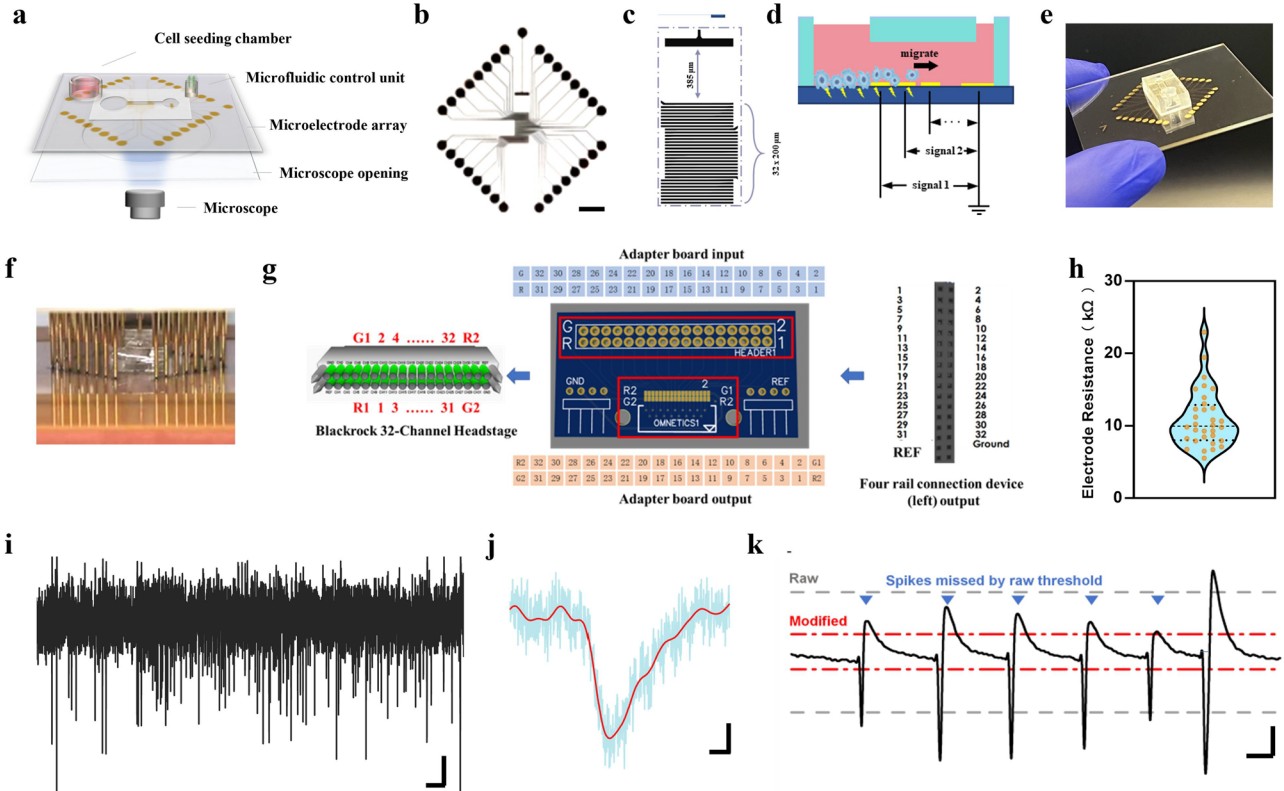

**Fig. 2 | Microfluidic Electrophysiological Platform for Real-Time Tumor-Neural Interaction Monitoring. a** The platform's monitoring module includes several core components: a cell seeding chamber, a microfluidic control unit to regulate fluid flow, a microelectrode array for electrical signal acquisition, a microscope opening for direct visualization, and a microscope. **b-c** Layout diagrams of the microelectrode array showing electrode configuration and spacing for optimal signal capture. Scale bar: 1 mm. **d-e** A microfluidic chip designed to initialize cell positioning and enable directional cell movement; (**d**) shows a schematic representation, while (**e**) provides an actual image of the setup. **f** A quick-clamp connector system allows for efficient, reliable connection to the microelectrode array.

**g** A custom adapter board interfaces the clamp-based connector with the front-end amplifier. **h** Impedance distribution map across the electrodes. ($n$ = 32 samples, The violin shapes depict the probability density of the data, the orange dots represent the raw data points, and the dashed lines indicate the interquartile range). **i** Real-time neural signals captured by the microelectrodes. Axis scale bar: 5 s, 40 µV. **j** Extracted neural peaks identified from neural signal data. Axis scale bar: 50 ms, 40 µV. **k** A modified thresholding algorithm designed to mitigate the effects of signal spikes, improving the accuracy of background noise estimation. Axis scale bar: 250 ms, 40 µV.

invasive behavior in glioma cells within specific microfluidic chips featuring distinctive neural signals, as evidenced by pronounced changes in cell distribution (Fig. 3e). Daily imaging revealed accelerated tumor movement in these hyper-invasive tumors, with cells traversing most electrodes by days 3-5. A histogram of daily cell positions (Fig. 3f) indicates that hyper-state invasive glioma migrates faster and more extensively over the same timeframe, underscoring enhanced motility and invasiveness. The temporal dynamics heatmap (Fig. 3g, h) further shows that tumor cells in the hyper-invasive state respond more rapidly, exhibiting faster and more widespread migration, whereas the normal-state tumor displays slower, less intense activity.

10,000 cells were seeded into the device (ratio of tumor cells to nerve cells is generally 1:1), and almost all seeded cells were retained within the device. Viability assessment demonstrated that cell survival rate remained above 80% following 7 days of culture (Fig. S4). To quantitatively assess the extent of tumor invasion, we introduced the Cell Migration Center (CMC) as a metric to define the overall migration of tumor populations within the microchannel. The CMC is calculated based on the weighted positions of cells, with cells closer to the end of the channel given greater weight. This method provides a comprehensive quantification of cell migration by integrating the spatial distribution of all migrating cells. The CMC allows for a more precise measurement of tumor invasion dynamics compared to individual cell tracking, as it reflects the collective behavior of the entire cell

population.

$$\widetilde{\mathrm{CMC}} = \frac{\sum_n^1 i \cdot CN_i \cdot d}{\sum_n^1 CN_i}$$

Where the variable "i" is defined as the electrode serial number, while "d" denotes the center-to-center distance between adjacent electrodes, "$n$" is the total number of electrodes, "$CN_i$" is the cell count near electrode "$I$". Compared to traditional single-cell tracking, this population-level analysis largely reduces data variability due to high intercellular differences in migration speed within the tumor population (Fig. S5). The quantitative CMC analysis (Fig. 3i) further corroborates our previous observations, showing a marked enhancement of glioma cell migration with neurons. The results indicate that glioma cells exhibit a baseline invasive behavior in the absence of neurons (defined as "Normal state", Fig. 3i). Notably, after the addition of neurons, the invasion rate of some tumor cells was observed to increase significantly (defined as "Hyper-Invasive", Fig. 3i). Hyper-invasive tumor cells exhibited rapid migration, with a sharp increase in CMC during the first 4 days, reaching the maximum migration distance by day 4, as some of them traversed the entire length of the microchannel.

**Unhijacked neural activity fails to induce tumor hyper-invasion**
To systematically resolve the distinct bioelectrical signatures of neurons and glioma cells in tumor microenvironments, we constructed a

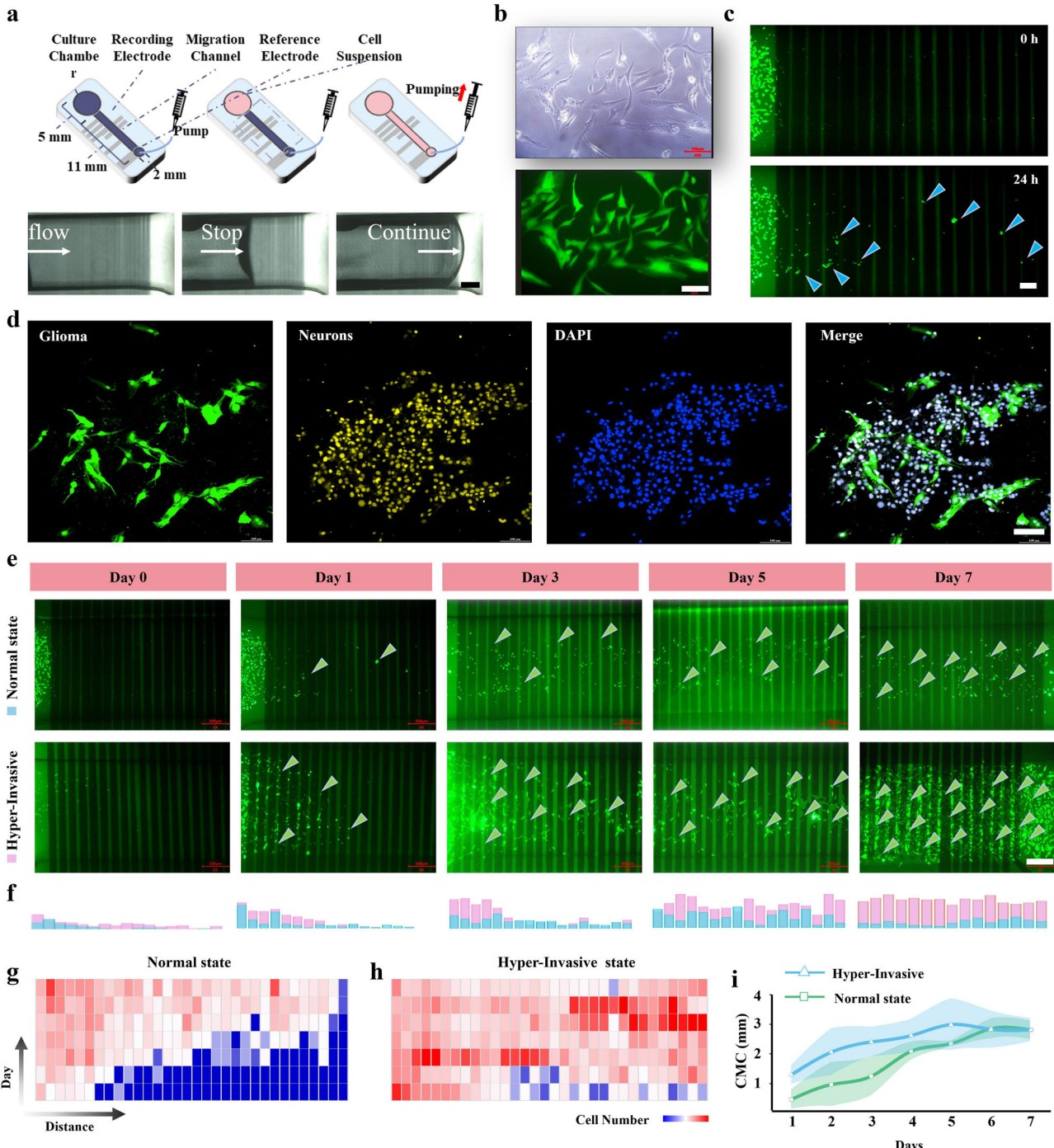

**Fig. 3 | Hyper-invasive behavior in brain tumors driven by specific neural networks. a** A pneumatic-based microfluidic system designed to consistently initiate glioma cell invasion within a neural environment, consisting of a main chamber, microchannel, air pump interface, and pump. **b** Glioma cells imaged under white light and GFP fluorescence. Scale bar: 100 μm. **c** Twenty-four hour migration of glioma cells within the microchannel. Scale bar: 100 μm. **d** Co-culture of glioma and neural cells, with glioma cells in green, neural cells in yellow (NeuN), and nuclei stained with DAPI in blue. Scale bar: 100 μm. **e** Upper panel shows glioma cell migration in the normal state over 7 days, while the lower panel shows hyper-invasive glioma migration in the presence of a neural network under specific stimulation. Scale bar: 500 μm. **f** Statistical bar plot of cell counts at each electrode location per day, with red indicating hyper-invasive glioma (with neurons) and blue representing glioma in the normal state (neuron-free). **g, h** Heatmaps illustrating glioma cell invasion over 7 days in the normal state (**g**) and hyper-invasive state (**h**). **i** Changes in the Cell Migration Center of glioma cells over 7 days under normal state (neuron-free) and hyper-invasive state (with neurons). *n* = 3 biological samples. CMC Cell Migration Center. Data are presented as mean values +/- SD.

standardized electrophysiological signal database under controlled conditions on the platform. We employed a microelectrode array (MEA) system to synchronously record transmembrane potential changes in both cell types (sampling rate: 30 kHz), with environmental noise suppressed using an adaptive Kalman filter. The raw signals after wavelet denoising revealed distinct characteristics: neuronal action potentials exhibited typical biphasic spike waveforms (duration: $1.2 \pm 0.3$ ms; peak-to-peak interval: $8.5 \pm 1.2$ ms), while tumor cells displayed prolonged multiphasic potential profiles (duration: 1 - 15.6 ms) with high variability in peak intervals (coefficient of variation: 32%).

These time-domain differences provide a critical basis for subsequent machine learning classification. In addition to conventional time-domain features (amplitude, slope, interval integral), we extracted energy distribution characteristics in the 0.1–5 kHz frequency band via Morlet wavelet transforms and derived nonlinear dynamic parameters of intrinsic mode functions (IMFs) using Hilbert-Huang transforms. After optimization with a recursive feature elimination (RFE) algorithm, 12 highly discriminative feature vectors were selected as inputs for a modified residual neural network (ResNet-18). This network achieved a classification accuracy of ~98% in cross-validation (Fig. S6, bottom right corner), demonstrating robust performance in distinguishing neuronal and glioma electrophysiological signatures.

To further explore the mechanisms underlying hyper-invasive behavior in gliomas within certain neural environments, we try to extracted electrophysiological signals from neurons and applied them directly to stimulate tumor cells. Our hypothesis proposed that certain type of neural spontaneous electrophysiological signals promotes glioma invasion by influencing intracellular biological signaling (Fig. 4a). Neuronal activity, essential for brain function, has been reported to modulate glioma cell migration via electrical interactions and calcium channel-dependent depolarization, which triggers calcium influx into adjacent tumor cells to regulate intracellular calcium levels and drive migratory behavior. Therefore, when glioma cells are exposed to these neural signals, they may exhibit changes in intracellular calcium levels that influence their migratory responses. Our calcium flux monitoring confirmed that spontaneous neural activity can indeed propagate calcium transients to adjacent glioma cells (Fig. 4b), establishing a pathway for electrophysiological crosstalk in the tumor microenvironment. Initially, Cell 1 exhibits initial firing, which propagates to Cell 2, subsequently propagating to cells 3 and 5. Upon simultaneous stimulation from Cells 1 and 3, Cell 4 (which seems with a higher activation threshold) is activated and engages in co-firing with Cells 1 and 3. It can be observed that Cell 4 itself exhibits reduced firing events, further supporting it may have a higher activation threshold and consequently lower excitability. Consequently, despite Cell 4's closer proximity to Cell1, the signal preferentially activates Cell 3, and only when convergent stimulation from Cell 1 and Cell 3 occurs does Cell 4 become activated and functionally incorporated into the form co-firing network. Additionally, signals from cells 2 and 5 showed no obvious co-firing with cell 1 (Fig. S7), indicating that these firing effects are locally transient and do not propagate further. Notably, we observed synchronized firing events between neurons and glioma cells (Fig. 4c), suggesting that glioma cells do not passively respond to neural signals but engage actively with this activity[3,5,51]. This coordination of calcium signals between neurons and tumor cells underscores the importance of neural-tumor electrophysiological interactions in influencing tumor invasion dynamics.

The extraction and application of neuronal stimulation signals enabled direct assessment of the influence of neural electrophysiological activity on glioma invasion. We captured and filtered real-time electrophysiological activity of neuronal networks who can promote tumor migration via our customized microelectrodes array, preserving key temporal and frequency-dependent information. The temporal domain analysis of the neural electrical signals (Fig. 4d) revealed two primary components: low-amplitude baseline fluctuations within ±10 μV and brief, high-amplitude peaks, which we define as "noise" and "code" respectively. This "neural codes" data retains the biological encoding necessary for evaluating its role in modulating tumor behavior, with the time sequence and shape of the signals being critical factors that likely contribute to decode biological information. We extracted stimulatory signals with specific patterns of neural activity that strongly promoted tumor invasion from 46519 neural electrophysiology signals. A detailed examination of these waveforms allowed us to classify and quantify their occurrence across all extracted signals, providing insights into the distribution of each category

and identify certain types of electrical stimuli being more effective in promoting tumor invasion. Through detailed waveform analysis, we identified and categorized a total of >1000 "neural codes" waveform types, and screened 295 frequently used codes (Fig. 4e–h, showcasing representative examples). Characterizing these "neural codes" is essential for understanding the electrophysiological mechanisms involved, particularly in identifying firing patterns associated with glioma invasion or other pathological processes, making them integral to bioelectrical signal interpretation.

To generate electrophysiologically valid stimulation signals, we trained a dual-branch LSTM model on 46,519 spatiotemporal neural event sequences encompassing peak types, inter-peak intervals, and sequential dependencies. At each timestep, the model processed three inputs: (1) one-hot encoded peak type, (2) continuous interval duration, and (3) historical signal context. The architecture employed two task-specific output branches: a softmax classifier for peak-type prediction and a ReLU-based regressor for interval estimation. Multi-task learning was implemented using a combined loss function (categorical cross-entropy + MSE), optimized via Adam (learning rate: 0.001) over 50 epochs with early stopping to prevent overfitting. The similarity between the simulated signal stimulation and nerve stimulation is shown in Fig. 4i (the relative occurrence frequency of code types with the most significant changes, the top 70 code types were selected with most of total variation). The application of neural spontaneous electrophysiological signals to glioma cells elicited a moderate enhancement of invasive behavior, confirming that the temporal dynamics and frequency modulation inherent in this neural activity has a stimulatory influence on tumor cell migration (Fig. 4j, k). However, this observed increase in invasion was less striking than the accelerated invasion patterns documented under hyper-invasive conditions (Fig. 4l), indicating that spontaneous neural network patterns alone may not fully explain the rapid invasion previously documented in hyper-invasive glioma. Building on prior research, we suspect that glioma cells could actively influence the electrophysiological patterns of the surrounding neural network, thereby promoting their own migratory behavior. This interaction may lead to the emergence of altered neural firing patterns.

## Hijacked neural signals drive hyper-invasive tumor behavior without direct neural engagement

While spontaneous neural network activity has been shown to promote glioma invasion, it alone does not fully account for the rapid invasion patterns observed in hyper-invasive gliomas. Meanwhile, another key observation from our research is that glioma cells not only respond to neural activity but also alter the spontaneous electrophysiological signals of surrounding neurons (Fig. 5a[3,5,51],). As illustrated in Fig. 5a, hijacked neural networks exhibited substantial alterations in signal frequency, duration, rise time, and decay time. These changes indicate that tumor cells are capable of modifying the waveform characteristics of neural signals. Furthermore, glioma cells were found to enhance the intensity of gamma (30-100 Hz) and theta (4−8 Hz) oscillations within the neural signals (Fig. 5b). Gamma waves are known to facilitate high-frequency neural oscillations essential for cognitive functions and inter-neuronal communication[52]. The observed increase in gamma wave activity suggests that glioma cells may enhance electrophysiological synchronization and promote high-frequency firing within neural networks. This hijacking of gamma wave synchronization is indicative of more coordinated and efficient signaling, which could, in turn, facilitate the invasive and metastatic capabilities of glioma cells. The elevated gamma wave activity correlated with increased glioma invasiveness, as high-frequency activity in this band is typically associated with elevated metabolic rates and enhanced cellular proliferation. In addition, the enhanced theta wave activity, generally linked to low-frequency neural processes such as spatial navigation and neuroplasticity, indicates a more profound glioma-neuron interaction. The increased theta wave activity likely

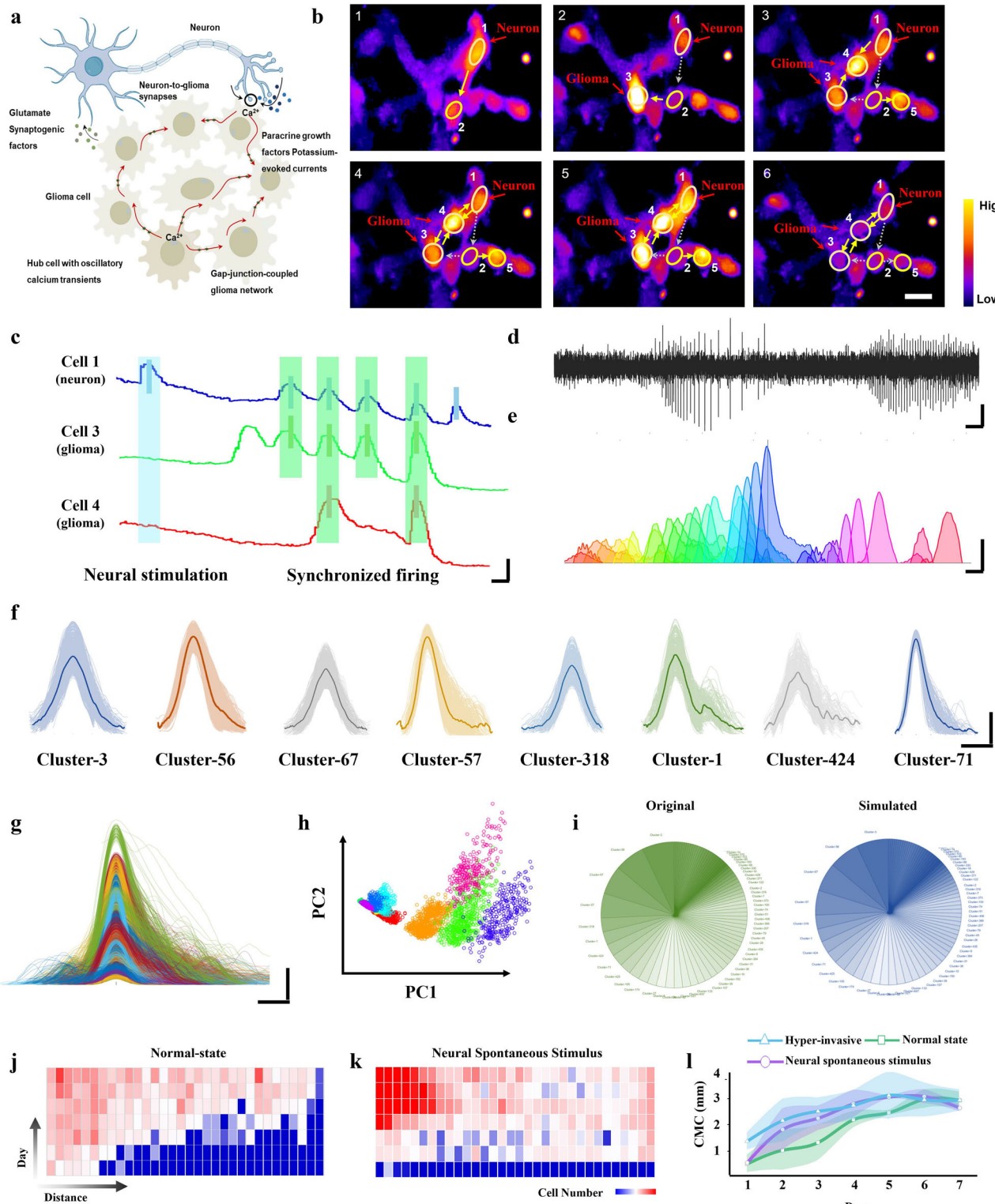

**Fig. 4 | Unhijacked neural activity fails to induce tumor hyper-invasion.**
**a** Schematic illustrating the electrophysiological interaction between neural cells and glioma cells, influencing tumor cell biological functions. **b** Cellular calcium signal transduction diagram. Scale bar, 20 μm. **c** Calcium ion intensity changes in three representative cells. Scale bar: 0.5 s, 0.2. **d** Baseline noise and peak codes within the neural signals. Axis scale bar: 5 s, 40 μV. **e-f** Reprehensive waveforms of commonly occurring neural signal codes. Axis scale bar: 100 ms, 100 μV. The solid line represents the template for this type of signal, and the colored area represents the range of signals of this type. **g**, **h** Classification display of several commonly used code waveforms. Axis scale bar: 100 ms, 100 μV. **i** Comparison of the distribution of the primary 70 code types between original (left) and simulated (right) neural signals. **j**, **k** Heatmaps showing glioma cell invasion over 7 days in the normal state (**j**) and neural spontaneous stimulus (**k**). **l** Changes in the Cell Migration Center of glioma cells over 7 days under normal state, hyper-invasive state, and neural spontaneous stimulus conditions. $n = 3$ biological samples. CMC Cell Migration Center. Data are presented as mean values +/- SD.

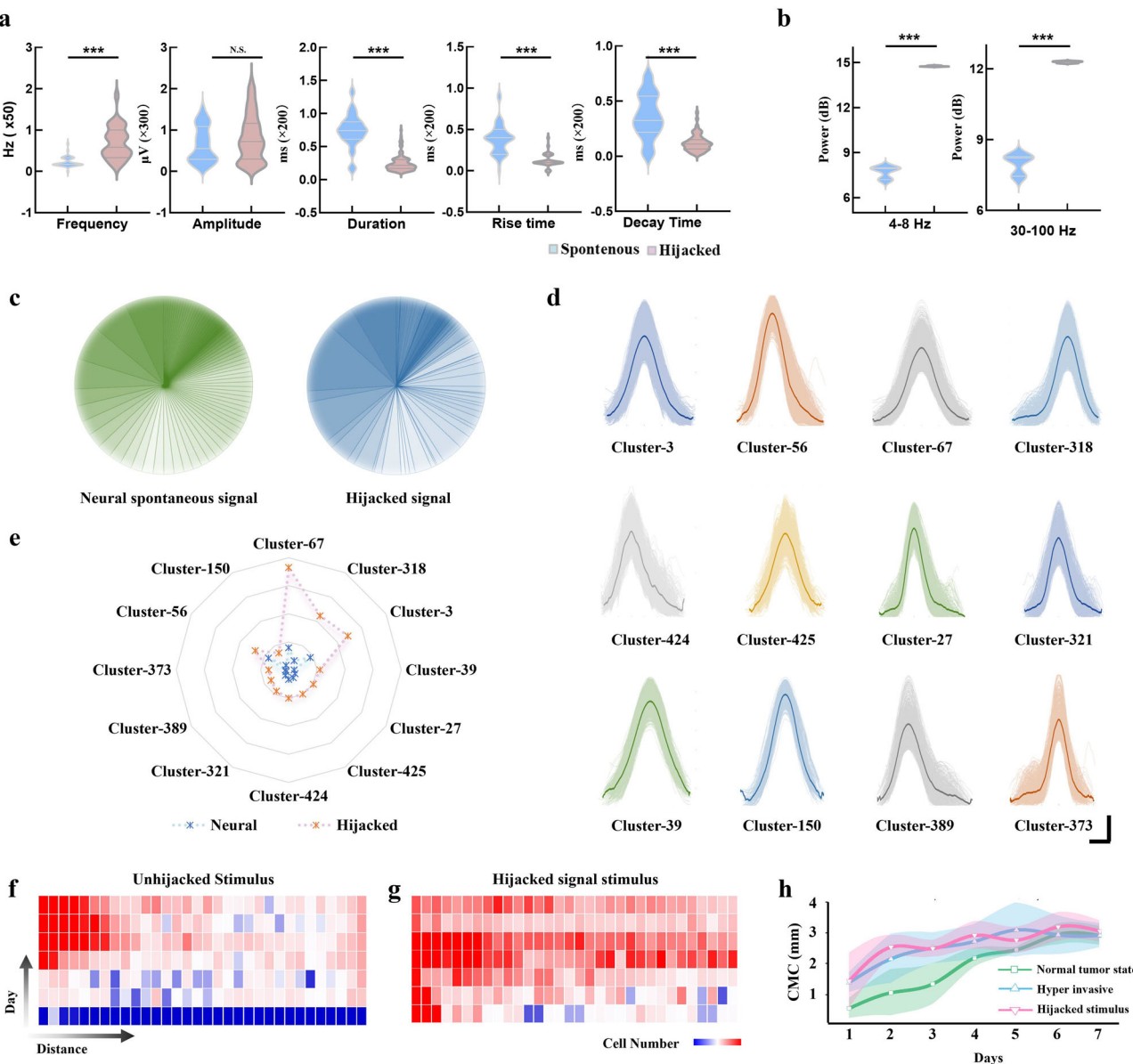

**Fig. 5 | Hijacked neural signals drive hyper-invasive tumor behavior without direct neural engagement. a** After being hijacked by hyper-invasive gliomas, neural signal properties, including frequency, duration, rise time, and decay time, undergo significant alterations (*n* = 3 biological samples). ***and n.s. indicate *p*-values < 0.001 and not significant, respectively, as determined by paired *T*-tests. Data are presented as mean values +/- SD. **b** Hijacking by hyper-invasive gliomas results in enhanced activity within the gamma (30–100 Hz) and theta (4–8 Hz) frequency bands of neural signals (*n* = 3 biological samples). *** indicate *p*-values < 0.001, as determined by paired *T*-tests. **c** The categories and proportions of encoded peaks within neural signals change significantly following hijacking by hyper-invasive gliomas. **d** Specific changes in key neural signal encoding peaks following hijacking by hyper-invasive gliomas. The solid line represents the template for this type of signal, and the colored area represents the range of signals of this type. **e-f** Heatmaps showing glioma cell invasion over 7 days in response to unhijacked (**e**) and hijacked (**f**) neural stimuli. **g** Changes in the Cell Migration Center of glioma cells over 7 days under normal tumor state, unhijacked stimulus, and hijacked stimulus. **h** Changes in the Cell Migration Center of glioma cells over 7 days under normal state, hyper-invasive state, and hijacked stimulus conditions. *n* = 3 biological samples. CMC Cell Migration Center. Data are presented as mean values +/- SD.

supports the establishment of communication pathways that allow glioma cells to navigate more effectively through the neural microenvironment, further promoting their invasive potential. To further dissect the biological basis of hyper-invasive behavior, we conducted molecular profiling and uncovered an enhanced invasion-associated transition marked by E-cadherin downregulation, Vimentin/Snail/Zeb1 upregulation, and concurrent dysregulation of EMT-related mRNAs (CDH1, VIM, SNAI1, ZEB1) (Fig. S8). These invasive molecular hallmarks validate the shift from a non-aggressive to a highly invasive phenotype. Based on this observation, we propose a hypothesis: glioma cells may hijack neural network signals to accelerate their migration (Fig. S9[3,5,51]).

To verify this hypothesis, we extracted the hijacked neural electrophysiological signals and applied them directly to glioma cells to determine whether they could induce the hyper-invasive state of tumor cells. These hijacked neural signals were captured and simulated as described previously for the neural signal extraction and simulation process. As shown in Fig. 5c, d, the neural signal encoding patterns in neurons also underwent significant changes under the influence of glioma cells. New waveform types emerged, while the proportion of certain pre-existing waveforms diminished. To investigate whether these hijacked signals could independently promote glioma invasion, we applied these altered electrophysiological signals

to glioma cells in isolation, without the influence of an intact neural network. Remarkably, we observed a significant increase in tumor invasion, comparable to that seen in hyper-invasive conditions (Fig. 5e–g). This finding confirms that glioma cells can indeed hijack neural signals to form new patterns that substantially enhance their invasive capacity. Notably, these effects were induced solely through electrical stimulation, without the involvement of any neural network-derived chemical mediators, suggesting that electrophysiological interactions may play a more fundamental role in promoting glioma invasion than previously understood. These results highlight the critical role of electrical activity in driving tumor progression, emphasizing that the electrophysiological properties of the tumor microenvironment are key facilitators of glioma invasive behavior.

## Discussion

This study sheds light on the intricate interplay between glioma cells and neural networks, emphasizing the role of hijacked neural electrophysiological signals in modulating tumor invasion. Our findings provide compelling evidence that glioma cells can actively utilize neural electrical signals to accelerate their invasion, highlighting a significant interplay between tumor biology and neural networks. Our platform overcomes the "signal ambiguity" problem in tumor electrophysiology by synergizing hardware design (zebra crossing electrodes) and computational analytics (spatiotemporal tagging). The ability to resolve "when and where" tumors electrically interact with neurons opens avenues for identifying bioelectric checkpoints in invasion pathways. The development of a pneumatic-based microfluidic electrode array system facilitated consistent tracking of glioma cell migration, allowing us to observe substantial cellular movement correlated with synchronized firing events of neurons and glioma cells.

Crucially, we found that spontaneous neural activity alone does not significantly boost glioma invasion; however, specific neural signals "entrained" by tumor cells significantly enhance their migration. This indicates that glioma cells engage in a bidirectional communication process that promotes tumor progression. Glioma cells altered the spontaneous electrophysiological signals of surrounding neurons, leading to synchronized firing that enhances intracellular calcium flux and creates a permissive microenvironment for invasion. Our analysis of neural signals captured via multi-channel microelectrode arrays identified specific waveform categories linked to enhanced tumor invasion, with deep learning models revealing critical temporal features that encode relevant biological information. The observed alterations in frequency and waveform characteristics, including increased gamma and theta wave activities, confirm that glioma cells exploit neural network dynamics to facilitate their migration.

We also further performed longitudinal quantification of GFP intensity to monitor glioma cell proliferation dynamics, which further validated that tumors exhibit not only increased invasiveness but also accelerated growth rates (Fig. S10). To verify whether identical bioelectrical signaling patterns induce comparable invasiveness across multiple cell lines, we introduced the human glioma cell line U251 to investigate consistent promotion of invasiveness by these signals across different lines. There is no significant difference between the U251 hijacked group and the Co-culture group. However, these two groups show differences from the other groups in the first 5 days, but no differences are observed on days 6–7. The results showed that while U87-associated bioelectrical signals moderately promoted invasion of U251, however, the signals derived from U251 cells themselves exhibit a stronger promotional effect on U251 invasiveness, highlighting variability in bioelectrical signaling patterns among cell lines (Fig. S11). We systematically evaluated glioma cell behavior across varying glioma-to-neuron ratios to determine a metric ratio in number of glioma cells versus neurons. As shown in Fig. S12, the enhanced hyper activity of the tumor increased with the increase in the proportion of glioma cells to neurons. When the ratio was 1:1, the enhanced hyper activity of the

tumor reached the highest level. Then, as the proportion of neurons further increased, this "hijacking" state gradually disappeared. This underscores the essential role of electrical activity in promoting glioma invasion, suggesting potential therapeutic strategies that target these electrophysiological interactions to disrupt glioma cell migration and improve patient outcomes.

## Methods

All procedures involving animals were reviewed and approved by the Animal Ethics Committee of Sun Yat-Sen University (SYSU-IACUC-2021-B0910).

### Fabrication of microfluidic electrode array chip

The microfluidic electrode array chip consists of a PDMS (poly-dimethylsiloxane)-based microchannel system combined with a gold-plated electrode array featuring 32 microelectrodes and integrated micro-gates. The PDMS microchannel was fabricated by casting PDMS pre-polymer (Sylogard 184 Silicone Elastomer Base, Dow Corning Corporation) at a 10:1 base-to-curing-agent ratio onto a micro-fabricated mold with the desired microstructure. The setup was then cured at 60 °C for 4 h. After curing, the PDMS was carefully peeled from the mold, preserving the channel features. The resulting micro-channel structure serves as both an invasion pathway for glioma cells and a conduit connecting the main cell loading chamber with the microfluidic air pressure control chamber. This chip allows precise regulation of air pressure within the channel, controlling fluid flow to suit experimental requirements. The main cell loading chamber, measuring 5 mm in diameter and 8 mm in height, is open to the atmosphere, enabling direct cell and culture medium loading. Inlet and outlet ports (5 mm and 2 mm in diameter, respectively) were punched at predefined positions to allow fluid control within the microchannel. An optical window was also integrated at the base of the platform to enable continuous microscopic observation of cell migration.

The measurement was performed in the customized device with electrode array on a micro-channel for tumor passing. The electrode array was designed in AutoCAD 2020 (Autodesk, Inc.) and fabricated via a standard photolithography-based lift-off process. Au electrodes (100 μm width × 3 mm length, 200 μm center-to-center spacing) were patterned by depositing a 100 nm-thick gold layer via magnetron sputtering (ISC-150, SuPro Instruments), with the reference electrode placed at the distal end of the channel, 4.6 mm from the last active electrode. Electrophysiological signals were amplified using an RHD-2000 pre-amplifier with a 1000x gain and sampled at 30 kHz to capture both slow local field potentials and rapid spiking neural activity. Calibration experiments include electrode resistance calibration, noise characterization, and inter-device consistency calibration. For each recording channel, baseline noise was measured in a cell-free "blank" culture chamber (identical to experimental chambers but without cells) over a 24-h period. To eliminate device-specific artifacts, three replicate MEA systems (Devices A, B, and C) were cross-validated using a standardized electrical signal source (sinusoidal wave: 100 μV amplitude, 10–500 Hz frequency range). For each device, two key metrics were assessed: signal amplitude deviation (<2% variation across devices after gain correction) and temporal synchronization, where time delays between input and recorded signals were adjusted via FPGA-based timestamp alignment (resulting in residual delays <1 ms). Devices failing to meet these criteria (e.g., >5% amplitude deviation) underwent hardware recalibration (e.g., amplifier gain adjustment) before experimental use. The filtering factor (3–5 × SD) is primarily grounded in classical neuroscience experience, with statistical properties of signal noise serving as a supplementary validation. The data acquisition system accommodates up to 512 channels, with synchronized data processing and transmission managed by an FPGA-based module. Noise reduction techniques, including adaptive filtering, were applied to improve signal clarity and reduce mean absolute

error. The chip was interfaced with external circuits via a quick-clamp vertical connector to facilitate rapid assembly. For cell culture applications, the assembled device was sterilized, and its surface was treated to promote cell adhesion through oxygen plasma cleaning (PT-BT10Plus, Sanhoptt Instruments Co., Ltd) for 300 s, followed by coatings of poly-l-lysine (A3890401, Thermo Fisher Scientific) and laminin (L2020, Sigma-Aldrich). The platform's design allows for long-term experiments, maintaining functionality for up to 2 weeks. Calibration of the system was performed using a standard signal generator (DG1002, Rigol Technologies), and validation experiments with cultured hippocampal neurons confirmed the platform's capacity to detect both spontaneous action potentials and local field potentials with high fidelity. This design, prioritizing sensitivity and low noise, enables precise, real-time monitoring of neural-glioma interactions during co-culture experiments.

## Characterization of microfluidic electrode array chip performance

Sealing integrity of the PDMS culture chamber and micro-grating electrode chip was validated through two sequential tests: (1) a 12 h leak test by injecting 200 μL deionized water into the microfluidic channel, confirming no fluid level changes, and (2) pneumatic pump-driven fluid manipulation monitored in real-time under a microscope (BDS400, Optec Instrument Co., Ltd). Electrode uniformity was verified by measuring resistance across all 32 electrodes (34465 A multimeter, Keysight). Signal transmission was tested using sinusoidal inputs (1 Hz/10 Hz, 2 mV; DG1002 generator, Rigol) with three replicates per frequency on three devices. Baseline noise was assessed by recording 10 second DMEM potential fluctuations (200 μL/channel) across three devices under identical conditions. All materials were validated for reproducibility through certified suppliers.

## Cell culture and immunofluorescence characterization

Glioblastoma cells (U87 MG, U251) were sourced from the American Type Culture Collection (ATCC, Manassas, VA, USA, HTB-14) and SSRCC (China, JY823) and cultured in DMEM/F12 (C11330500BT, Gibco), supplemented with 10% fetal bovine serum (FBS; 10270-106, Gibco), 2.5 mmol/L L-glutamine (25030081, Thermo Fisher Scientific), and 100 U/mL penicillin-streptomycin (15140122, Thermo Fisher Scientific). For real-time tracking, glioblastoma cells were transfected with a GFP expression plasmid using Lipofectamine 2000 (Thermo Fisher Scientific) according to the manufacturer's protocol. E18 embryos collected from one pregnant female Sprague-Dawley rat (Crl:CD[SD] from Charles River, 8-10 weeks old) were euthanized via $CO_2$ asphyxiation, embryos decapitated, and brains extracted into ice-cold HBSS (without $Ca^{2+}/Mg^{2+}$) with 10 mM HEPES (pH 7.4); hippocampi were microdissected under a stereomicroscope and dissociated via 15 min 37 °C incubation in papin, followed by gentle trituration; the cell suspension was centrifuged, resuspended in Neurobasal-A medium (supplemented with 2% B27, 0.5 mM L-glutamine, and 100 U/mL penicillin/streptomycin) for use. Notably, even though GBM does not originate in the hippocampus, evidences have shown that hippocampal neurons do engage in functional interactions with GBM cells[1,35], particularly through conserved synaptic and excitatory signaling pathways. Furthermore, hippocampal neural networks represent a uniquely well-characterized model in neurobiology: they encompass both glutamatergic (excitatory) and GABAergic (inhibitory) neuronal phenotypes, and their robust synaptic functionality and high experimental reproducibility have been consistently validated across decades of studies.

Cells were plated at ~70% confluency and transfected with a Lipofectamine 2000-DNA complex. Following transfection, cells were incubated for 24 h to achieve sufficient GFP expression, facilitating visualization during migration studies. Hippocampal neurons were isolated from E18 rat embryos and cultured in Neurobasal medium

(21103049, Thermo Fisher Scientific) supplemented with B27 (17504044, Thermo Fisher Scientific) and L-glutamine (25030081, Thermo Fisher Scientific). All cultures were maintained at 37 °C in a humidified environment with 5% $CO_2$ for optimal growth. Prior to experiments, the microfluidic electrode array chip was coated with an adhesion-promoting layer of 1 mg/mL Poly-l-lysine (A3890401, Thermo Fisher Scientific) and 0.5 mg/mL Laminin (L2020, Sigma-Aldrich) to enhance cell attachment to the electrode array. Cells (20,000) were then introduced into the microfluidic channels, and glioblastoma cell migration was tracked under both brightfield and GFP fluorescence modes using a microscope, facilitated by the platform's optical access. The culture medium within the microfluidic chip was regularly replaced to sustain cell activity, and cell viability was monitored by periodic observation of cellular morphology and electrophysiological signals to ensure normal cellular function.

To visualize neural markers in the co-culture, immunofluorescence staining was performed. Cells were washed twice with phosphate-buffered saline (PBS, BL601A, Biosharp) and fixed in 4% paraformaldehyde (158127, Sigma-Aldrich) for 15 min at 4 °C. After fixation, cells were rinsed three times with PBS, and non-specific binding was blocked with 5% bovine serum albumin (BSA, A3294, Sigma-Aldrich). Cells were incubated overnight at 4 °C with the primary antibody NeuN (1:500; 12943 T, Cell Signaling) in 5% BSA. Following primary antibody incubation, cells were washed five times with PBS, then incubated with a Cy3-conjugated secondary antibody (goat anti-rabbit, 1:500; GB21303, Servicebio) in 5% BSA for 4 h at room temperature in the dark. After a final series of five PBS washes, cells were counterstained with DAPI (C1002, Beyotime) for 10 min to visualize nuclei, followed by five additional PBS rinses. Immunofluorescence imaging was performed using a Nikon AX confocal fluorescence microscope (Nikon Corporation), with excitation wavelengths of 488 nm (for GFP), 550 nm (for Cy3), and 350 nm (for DAPI). This imaging approach enabled high-resolution brightfield and fluorescence visualization of glioblastoma and neural cells in co-culture.

## Calcium imaging

Calcium transients in co-cultured glioma and neuronal cells were visualized using calcium-sensitive dyes, Fluo-8 AM (ab142773, Abcam). Cells were incubated with 10 μM Fluo-8 AM dye for 30 min at 37 °C in the dark. After incubation, cells were washed three times with phosphate-buffered saline (PBS, BL601A, Biosharp) to remove excess dye, followed by the addition of fresh phenol red-free culture medium (DMEM/F-12, no phenol red, 21041025, Gibco) to facilitate imaging. Fluorescence imaging was performed using a Nikon AX confocal microscope (Nikon Corporation) to capture intracellular calcium level changes. Fluctuations were recorded as variations in fluorescence intensity within glioma cells adjacent to the neuron co-culture chamber, enabling visualization of calcium transient responses to neuronal electrophysiological activity. For quantitative analysis, calcium oscillations in both cell types were recorded over a 5 min period. Fluorescence intensity changes were analyzed in ImageJ software (NIH, Bethesda, MD, USA) to assess the temporal dynamics of calcium transients triggered by neuronal signals. Results were plotted to compare intracellular calcium level variations between glioma and neuronal cells under co-culture conditions.

## Multi-channel electrode array recording and waveform classification

The electrophysiological activity of neurons was recorded using a custom-designed multi-channel microelectrode array (MEA) system after 10–14 days of culture to ensure mature synaptic connectivity. For each recording session, a sterile micro cover was positioned over the cell culture chamber to maintain a sterile environment throughout the experiment. After connecting the MEA chip to the preamplifier, the system was allowed a 5 min stabilization period before recording.

Once the environment reached stability, electrophysiological signals were captured for a duration of 610 s across 32 channels. Signal detection in the MEA system relied on the cell-electrode coupling model, as represented in Fig. S13. Neurons adhered to the electrode surface, separated by a nanoscale gap of culture medium (typically 40–100 nm). Cells adhered to the sensing electrode, separated by a nanoscale culture medium gap (typically 40–100 nm), which created an equivalent circuit of the cell-electrode interface. This interface was modeled with the cell membrane as a parallel combination of a capacitor and a resistor, representing the phospholipid bilayer capacitance and ion channel impedance, respectively. The impedance between the cell membrane and electrode surface, termed $R_{seal}$, represented the gap resistance. Signal detection was optimized by adjusting the coupling ratio ($V_{in}/V_s$), where $V_{in}$ denotes the input voltage to the amplifier and $V_s$ represents the source voltage generated by ionic fluctuations at the electrode interface.

Neural signals were captured in real time by the MEA system, analyzed for both temporal and frequency characteristics, including low-amplitude fluctuations and high-amplitude spike events. Data were digitized and processed via an FPGA-based acquisition module. Peak waveforms were extracted, initially categorized by amplitude, frequency, and temporal features, and stored in binary format. Raw signals were filtered using a second-order Butterworth low-pass filter with a 3 kHz cutoff to mitigate noise, and non-causal zero-phase filtering was applied to prevent phase distortion. A Savitzky-Golay FIR smoothing filter (polynomial order 5, frame length 501) was subsequently employed to improve signal clarity. A threshold adjustment algorithm was employed to estimate distribution parameters (mean and standard deviation) via maximum likelihood estimation, eliminating artifacts and outliers. This approach ensured accurate noise distribution assessment, thereby improving the reliability of recorded signal analyses.

Neural signal decoding was conducted using an unsupervised waveform classification process, constructing a comprehensive waveform database without predefined categories. Labeled training pool was constructed manually. Initial waveforms were aggregated into a waveform library, where the first waveform served as the standard template for categorization. Common-mode noise was firstly eliminated by subtracting overall signal fluctuations from each recording site. Next, bidirectional application of zero-phase Butterworth filtering (1 - 3000 Hz) was applied to mitigate phase distortion. The active signals were then extracted and peak-aligned to time 0, calibrating their peak amplitudes to the 0-time reference point, before proceeding to waveform similarity analysis. Waveform similarity comparison and classification were performed by digit-by-digit comparison from the 0-time point outward, with label consistency evaluated against a predefined threshold (30%). Waveforms with ≥90% label consistency were grouped into a fixed and unique neural code type. Each new waveform was cross-checked against existing template classes in the neural code library, and unmatched waveforms were designated as new neural code types.

## Glioma invasion assay under neural stimulation

The glioma invasion assay in response to neural stimulation was conducted using a custom-designed microfluidic co-culture chip. Neural spontaneous electrophysiological signals were collected from a dedicated device containing pure neural networks (without tumor cells), and required signals were then applied to a separate device with pure tumor cells via the microelectrodes at the bottom of the microfluidic chamber. The electrical stimulation was delivered via integrated microelectrodes within the microfluidic device (recording function of the stimulated microelectrodes was paused during stimulation). The stimulation duration was set to last throughout the entire experimental period, during which the stimulation was continuously applied to the tumor cells to simulate neuron-tumor interactions in co-culture.

Prior to cell seeding, the chip surface was coated with 1 mg/mL Poly-l-lysine (A3890401, Thermo Fisher Scientific) and 0.5 mg/mL Laminin (L2020, Sigma-Aldrich) to enhance cell adhesion. Glioma cell migration was monitored over a 7 day period using a microscope (BDS400, Optec Instrument Co., Ltd) with 488 nm excitation to detect GFP-expressing cells. Daily imaging was performed to capture migration progress. Quantitative migration metrics, including invasion distance and the Cell Migration Center (CMC), were analyzed using ImageJ software (NIH, Bethesda, MD, USA). The CMC was introduced as a comprehensive measure of tumor invasion, accounting for the spatial distribution of migrating cells within the microchannel. CMC was calculated using the following formula:

$$\widetilde{CMC} = \frac{\sum_n^1 i \cdot CN_i \cdot d}{\sum_n^1 CN_i}$$

where $i$ represents the electrode serial number, $n$ is the total number of electrodes, $CN_i$ denotes the cell count near electrode $i$, and $d$ is the center-to-center distance between electrodes. Waveform events were extracted, and both temporal and frequency data from neural signals were examined to explore potential interactions between glioma cell invasion and neural activity.

To generate stimulation signals containing neural encoding with valid electrophysiological information, we extracted spatiotemporal sequences of encoded neural events (the types of encoded peaks, their inter-peak time intervals, and the types of subsequent peaks), and then processed these data using a deep learning model (Long Short-Term Memory, LSTM). This model was trained on 46519 extracted signals, capturing both the temporal dependencies and the distribution of waveform types. The LSTM network consisted of two layers, each containing 128 hidden units, enabling the model to learn both short-term and long-term dependencies in the signal data. At each time step, the model received three key features: a one-hot encoded vector representing the neural peak type, the time interval between consecutive peaks as a continuous feature, also historical signal sequences for contextual learning. After passing through the LSTM layers, two fully connected layers were added to process the temporal features further. The output layer was split into two branches: one branch predicted the peak type using a softmax activation function, and the other predicted the time interval between peaks using a ReLU activation function. The model was trained using a multi-task learning approach, with a combined loss function that incorporated categorical cross-entropy for peak type prediction and mean squared error (MSE) for time interval prediction. The model was optimized using the Adam optimizer with a learning rate of 0.001, and training was carried out for 50 epochs, employing early stopping to avoid overfitting.

Once trained, the model was used to generate simulated electrophysiological signals. This process began with a randomly selected seed from the dataset, and the model sequentially predicted the peak types and time intervals for each successive signal point. To ensure that the generated signals matched the statistical distribution of the original data, a statistical control mechanism was applied. This mechanism adjusted the predicted peak category distributions and timing based on the observed characteristics of the original dataset. The generation process continued until the simulated signal contained at least 70 occurrences of common code types, ensuring that the distribution of each code type within the signal matched the overall proportions observed in the original dataset. The generated signals were then evaluated using statistical tests, such as the Chi-square test or Kolmogorov-Smirnov (K-S) test, to compare the temporal structure, waveform distribution, and timing patterns between the generated and real signals. Additionally, visual comparisons were made to assess how closely the simulated data resembled the original experimental data. This approach provided a robust framework for simulating

neural signals that influence glioma invasion, helping to elucidate the role of neural-tumor crosstalk in cancer progression.

## Extraction and application of hijacked neural electrophysiological signals

To extract hijacked neural electrophysiological signals, chips displaying a hyper-invasive state were screened and peak sorting was employed to isolate hijacked neural activity within neuron-glioma co-cultures, focusing on the temporal waveform characteristics of these signals. A total of 127,706 electrophysiological signals were initially extracted, followed by spectral analysis to identify frequency-domain alterations and synchronization patterns indicative of signal hijacking by glioma cells. Further analysis focused on characterizing waveform properties specific to the hijacked signals. To help the model capture the temporal patterns specific to each waveform type, each waveform category is embedded as a low-dimensional feature vector using an embedding layer, which provides a learnable representation for each category and allows the model to better understand inter-category relationships and temporal dependencies. The model architecture consists of a two-layer LSTM network, each layer containing 128 hidden units. This structure is designed to capture both local and global temporal dependencies within the signal. A dropout rate of 0.3 is applied between layers to reduce overfitting. The model is trained using the Adam optimizer with a learning rate of 0.001, and a multi-task output layer is employed. One output branch predicts the next time step's signal value, applying a ReLU activation to ensure non-negativity, while the other branch estimates the distribution of waveform categories using a Softmax activation. This design allows the model to learn both the dynamic sequence patterns and the probabilistic distribution of waveform categories over time. For training, the model uses a combined loss function: mean squared error (MSE) for signal prediction and categorical cross-entropy for waveform category distribution. This multi-task approach enables the model to simultaneously learn temporal dependencies and categorical distributions, making it capable of generating realistic waveform sequences.

During the generation phase, the model starts from an initial randomly selected seed from the dataset and generates successive waveform points. To ensure the generated data matches the original data's category proportions and temporal structure, a statistical control mechanism adjusts the predicted category distributions and timing based on the original dataset's distributional characteristics. Finally, the generated signals are evaluated using statistical tests (e.g., chi-square or K-S tests) and visual comparisons to confirm that the simulated data closely resembles real signals in terms of temporal structure, waveform distribution, and timing patterns. For application, glioma cells were exposed to electrical stimulation patterns that mimicked the hijacked signals, including adjusted waveform proportions and preserved gamma (30–100 Hz) and theta (4–8 Hz) frequency activities. A custom stimulation protocol, informed by the GRU model, was implemented to deliver pulse trains at the identified hijacked signal frequencies and durations observed in co-culture conditions. Following stimulation, invasion assays quantified glioma cell migration and invasion to evaluate the cells' response to the applied hijacked signals.

## Immunoblotting

Cells were lysed in RIPA buffer (PC101, Epizyme) supplemented with protease inhibitors (GRF101, Epizyme). Lysates were centrifuged at 12,000 rpm for 30 min, and the supernatants were collected. Equal amounts of protein were separated by SDS−PAGE and transferred onto polyvinylidene difluoride (PVDF) membranes (E802-01, Vazyme). Membranes were blocked with 5% non-fat milk (3191345, BD) in TBST for 1 h at room temperature, followed by incubation with the indicated primary antibodies overnight at 4 °C with gentle shaking. After washing, membranes were incubated with HRP-conjugated secondary antibodies (1:10,000; Abbkine, A21020, A21010) for 1 h at room temperature. Protein bands were visualized using an enhanced chemiluminescence (ECL) detection system (Yeasen) and imaged using a BIO-RAD imaging system. Primary antibodies used in this study included: E-Cadherin (1:1,000, CST, 3195), Vimentin (1:1,000, CST, 5741), Snail (1:1,000, CST, 3879), ZEB1 (1:1,000, CST, 70512), and GAPDH (1:10,000, Abbkine, A01020).

## Quantitative RT-PCR

Total RNA from cells was extracted using the FastPure Cell/Tissue Total RNA Isolation Kit V2 (RC112-01, Vazyme). According to the manufacturer's instructions, cDNA was synthesized using the Prime-Script RT reagent kit with gDNA Eraser (AG11706, Accurate Biology). Real-time quantitative PCR was performed using the SYBR Green Premix Pro Taq HS qPCR Kit (AG11701, Accurate Biology) according to the manufacturer's instructions. Relative gene expression levels were normalized to GAPDH expression and calculated using the $2^{-\Delta\Delta CT}$ method. The primers used in this study were synthesized by Tsingke Biotechnology (China) and are listed in Table S1.

## Statistics and reproducibility

All quantitative data are presented as mean ± standard error of the mean (SEM), unless otherwise stated. Cellular calcium signal processing was performed using Fiji (ImageJ; National Institutes of Health, Bethesda, MD, USA). Figures were generated using microsoft Office 365 (Microsoft Corporation, Redmond, WA, USA) and OriginLab 2023 (OriginLab Corporation, Northampton, MA, USA). In general, data were tested for normal distribution by Kolmogorov-Smirnov normality test and analyzed accordingly by unpaired two-tailed $t$-test were used to perform pairwise comparisons between two groups. Each experiment (e.g., migration assay, calcium imaging, or electrophysiological recording) was independently repeated at least three times with consistent results. Details of the data values, sample sizes and statistical measurements are provided in the figure legends.

## Reporting summary

Further information on research design is available in the Nature Portfolio Reporting Summary linked to this article.

# Data availability

All data needed to evaluate the conclusions in the paper are present in the paper and/or the Supplementary Materials. Source data are provided with this paper.

# Code availability

The core codes and reference libraries for biosignal classification are available at https://github.com/BingzheXu/BioDecoder. The BioDecoder platform enables classification and interpretation of biosignals from various biological sources. Researchers can upload extracted signal peaks (merged_peak_library.mat), obtain standardized classifications (Classified_Peaks.mat), and explore the biological significance of waveforms based on integrated research data.

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

## Acknowledgements

This work was supported by the National Natural Science Foundation of China (Grant number: 32101160), Foundation of Guangdong Provincial Key Laboratory of Sensor Technology and Biomedical Instrument (Grant number: 2020B1212060077).

## Author contributions

B. X. conceived and supervised the project, decided research methodologies and prepared the manuscript; T. X. conducted experiments and contributed to figure preparation; Y. J. helped to revise manuscript. X. Z. assisted with cell-based experiments and signal analysis; K. S. contributed to the development and optimization of the electrophysiological platform. J. L., J. R., J. H., provided experimental support. C. L., Z. Y., H. J., B. Z., J. L. offered clinical guidance. N. L., provided technical advice and equipment support.

## Competing interests

The authors declare no competing interests.
