## [Transparent Peer Review file · Nature Communications]

A Machine Learning-Driven Electrophysiological Platform for Real-Time Tumor-Neural Interaction Analysis and Modulation

Corresponding Author: Dr Bingzhe Xu

Version 0:

Reviewer comments:

Reviewer #1

(Remarks to the Author)

In this manuscript the authors develop a novel platform for measuring neural and glioma cell electrical signal patterns while utilizing machine learning algorithms to be able to reliably distinguish between neural-specific and glioma-specific bioelectrical signal patterns. The authors study glioma invasion in vitro with this platform and correlate tumor invasiveness to the specific electrical signal patterns being recorded by their platform. They find that specific neural networks can promote hyper-invasive behavior of brain tumor cells. They first applied neural-specific bioelectrical signals to glioma cells and found a modest increase in invasiveness although not the hyper-invasive phenotype. Next, they stimulated glioma cells with bioelectrical signal pattern of hijacked neural networks where glioma cells had modified the waveform characteristics of the bioelectrical signal patterns of neural cells. They found this glioma-altered neural network signal significantly accelerated tumor cell migration. Interestingly, the electrophysiological signal alone without any neurons was enough to increase tumor invasion.

This work is of significance to the field of glioblastoma pathophysiology and the interplay of neuron-glioma circuits. The authors do a good job of describing the current state of the literature in this field in their introduction and their work in this study appropriately expands upon our current knowledge. The platform utilized by the authors in this study is novel and provides key insights into how glioma cells alter the bioelectrical signal pattern of neurons and how these electrical circuits promote glioma invasion. The results and conclusions presented by the authors in this manuscript logically make sense within the context of the current state of the literature and our understanding of how neuron-glioma circuits function. The authors do a good job supporting their claims and conclusions with the experiments and figures provided in this manuscript. The methodology appears sound and definitely the correlation of the authors' in vitro experiments and immunofluorescence data to the bioelectrical signals recorded is convincing. A reviewer with a stronger background on computational biology should comment on the validity of the authors' machine learning algorithm calculations. This work meets the expected standards in this field.

In general, there is sufficient details provided in the methods for the work to be reproduced, please see suggestions below for some points for improvement. Overall, this is a well-written manuscript that is clear to understand. The author's present some very interesting results that are convincing based on the experiments completed and also make sense within the larger context of the established literature in this field.

Suggestions:

- In the introduction, the authors could clearly state their hypothesis towards the end of this section. The authors talk about on lines 113-114 that mimicking electrophysiological signals can replicate tumor phenotype in vitro. However, they could state their hypothesis here which they actually talk about in their results and discussion section. Something along the lines of "Given our understanding that neuron-glioma interactions are bi-directional, we hypothesize that the electrophysiological signals produced by the interaction between these two cells types will have a greater effect on tumor invasiveness than the electrophysiological signals produced by neural cells alone" or something to that effect which seems to be essentially the central hypothesis of this paper. Presenting that in the introduction could help the reader better frame the results/discussion sections.
- One experiment to potentially consider is testing different cell lines of glioma cells to determine if there are any differences between cell lines or if the same bioelectrical signaling patterns induce a similar degree of invasiveness across multiple cell lines.
- Additionally, it may not be too challenging to also look at glioma proliferation especially if all of the glioma cells are tagged with GFP. Measuring change in GFP intensity over time could be used to determine whether the tumors are growing faster

as well as being more invasive as I suspect they may be.

- Authors should provide the glioblastoma cell line or cell lines utilized in the co-culture experiments as it is not immediately clear which cell lines were used and whether these cell lines are derived from mouse, rat, or human.
- For the experiment described in figure 4-5, the authors state glioma cells were exposed to electrical stimulation patterns but it is unclear how this electrical stimulation is applied to the tumor cells. More details on how the electrical stimulation was applied to the glioma cells could help clarify the methodology. Also, any information regarding duration of stimulation application would be helpful as well and provide further context for this experiment. For instance, was the stimulus provided for a total time of 30 minutes or a couple of hours or did it vary?

Reviewer #2

(Remarks to the Author)

The authors developed an MEA + machine learning platform to study the interaction between GBM cells and neurons. This new platform can answer important questions in cancer neuroscience. The interesting findings from this novel system are that as long as the authors mimic delivering the neuronal electrical signal to the sole glioma culture, they observed a similar glioma migration-promoting effect as seen in the neuron-glioma co-culture, suggesting that neuronal activity-dependent factors may not be necessary in this phenomenon; however, the biological model system needs to be further developed and validated to support the conclusion.

- It is unclear which GBM cell line the author used. Detailed information is not included in the methods. Regardless, using only one GBM cell line does not reflect the high degree of heterogeneity present in GBM. Therefore, multiple cell lines, particularly human cell lines, should be tested in this in vitro system.
- It is unclear whether "normal state" and "hyper-invasive state" describe the same GBM cell line, just different behaviors in the MEA chambers. Additionally, what criteria are used to define these two states?
- The authors claim that their system tests GBM cell invasion, which is debatable. The system forces glioma cells to enter the neuron-rich area. Therefore, GBM cell mechanical sensing and their attachment to the bottom both contribute to their migration toward the neurons. The authors should utilize molecular assays to distinguish between cells in the "normal state" and the "hyper-invasive state," and further evaluate whether the "hyper-invasive state" actually exhibits enriched molecular signatures in cell migration and invasion.
- Several sentences (lines 262-266, 310-312, 330-331) appeared as if they were novel findings by the authors; however, they had already been shown previously by multiple groups in vivo (Monje, Winkler, Deneen labs). The claims need to be softened, and prior studies need to be cited and discussed.
- In the methods, it's unclear how hippocampal neurons were isolated from E18 rat embryos. The system is somewhat unusual because GBM rarely forms in the hippocampus. What neuron subtypes are primarily present in this culture?

Minor comments:

- In many places, there is no space before the first parenthesis of the reference.
- Lines 316-318 need citation.
- line 300, not sure why "pronounced" is used.
- Line 303, not sure what "hat glioma cells" means.
- Fig. 3d should specify which neuron marker is used in the image and/or legend

Reviewer #3

(Remarks to the Author)

The manuscript deals with a proposed machine learning-based electrophysiological platform that combines custom microfluidics with real-time decoding of neural-tumor signal interactions. The manuscript claims that glioma cells do not passively respond to neural activity but instead selectively hijack specific neural electrical patterns. The authors claim that this selective entrainment leads to synchronized firing between neurons and tumor cells, which in turn drives hyper-invasive tumor behavior. Furthermore, the authors demonstrate that direct stimulation of glioma cells with these hijacked patterns is sufficient to induce aggressive invasion. The manuscript presents this platform as a novel tool for dissecting dynamic tumor-neural communication and suggests its potential for translational applications targeting bioelectrical signaling in glioma progression.

The manuscript is well written and presents a wide variety of experiments and data analytics endeavors. However, it does not typically provide full contextualization and details about the experiments being performed. The developed platform is quite complete, and it is shown that it is capable to precisely acquire signals from glioma and neurons. Despite this, I believe

there are some experiments missing regarding the assessment of cell viability, cell-cell interaction and even the claimed effect on tumor cells, as, for example, it was not shown if glioma cells would move across the device even if neurons were not present. Moreover, certain concepts like what constitutes “hijacking” or why certain waveforms might be more relevant than others are not properly explained. Because of this, I believe that further experiments and results must be added in order to reach publishable status, as well as editing on the text to make many of the results presented more clearly understandable. Please, find below a list of comments to be addressed in the next version of the manuscript.

Comments:

- Page 3, Line 7-11: During Introduction, more context should be provided on the relationship between cancer cell exploitations of ion channels and their “high proliferative rate, migratory behavior, and invasiveness(14)”. Without further contextualization, it does not become clear why, for example, mutations in ion channels of 90% of glioblastoma should be related to aggressiveness.
- Page 3, Line 21-22: When stating that “excitatory postsynaptic currents (EPSCs), (lead) to glioma cell membrane depolarization and promoting tumor proliferation”, it would help readers if more details about glioma as a disease, its onset, how does it typically expand and what are some of its key proliferative mechanisms, for example.
- Page 5, Line 22-25 (and Method section): Can the authors provide further information regarding the culturing conditions, specifically, viability assessment along experimental time and typical numbers of cultured cells? Also, did the authors assess the level of adhesion of cells to the coated electrodes, i.e., was it assessed if all cells or only a small fraction of the seeded cells stayed within the device?
- Page 7 – Can the authors provide more information regarding how measurements were performed and how the electrode array was designed? Also, further details on calibration experiments or efforts to understand how the signals might vary under different conditions would be valuable. Moreover, on the filtering criteria provided, how is the factor applied determined?
- Page 8 – Can the authors report on control experiments where the neurons were not initially present on the chamber before glioma cells are added? It would be critical to confirm if the presence of the neurons within the chamber has an effect on the movement observed.
- Page 8 – It is not clear to me what constitutes the “hyper-invasive behavior”. How do the authors identify this behavior? What metrics, signals or other, are used to classify them as such?
- Page 8 – Can the authors comment on the ratio of glioma cells that started movement versus the population that stayed within the input sample? Figure 3 does not make it clear if all the cells or only a small fraction actually initiated movement across the device.
- Page 8/9 – Can the authors comment on the observed movement of glioma cells, i.e., if their movement followed any specific trend, sequence, morphological change or any other behavior that could also be important to highlight?
- Page 8/9 – During the 7 day experiment, how was viability of cells ensured if the liquid within the chamber was stopped in order to ensure no further movement induced into the channel?
- Page 9 – It is confusing to use “i” as a variable for both the electrode serial number and the center-to-center distance of electrodes. Also, it is not clear the difference between “i” and “d” on that regard? Moreover, the authors should provide, even if in supplementary and for a small set of experiments, a comparison between the classical approach highlighted of individual cell tracking and their proposed approach, in order to showcase the differences and potential advantages of the proposed method.
- Page 10 – Figure 4B – It is not very clear what is a neuron versus glioma cell in the images provided. For example, it seems that the Calcium trigger goes from Neuron (1) to Glioma (2) around the Glioma (3), while at the same time there are other cells (?)/locations with high signal as well that are not annotated or receive any comments. I would ask the authors to better annotate the Figure, as well as providing a better discussion of the phenomena being observed in it.
- Page 11, Line 2 – Can the authors comment why the synchronized firing might have went from Neuron (1) to Glioma (2) to (3), instead of (1) to (3), as Glioma (3) is much closer to Neuron (1)? Also, why does Glioma (3) have less firing events compared with the other cells? And can the authors analyze and provide the data measured at other cells that were found surrounding these 3 cells? Is this firing effect short and local, or does it propagate further into other cells?
- Page 11 – Line 22 – The authors must properly explain how the “neural codes” were identified and distinguished into more than 1000 types. It is not clear what metrics were used to compare them or how they were evaluated. Also, how was comparison across different experiments and observed glioma behaviors for ex?
- Page 12 – Figure 4i – I am unfamiliar with the type of plot utilized to compare the simulated signal with the nerve stimulation signal. Can the authors comment on what is being evaluated and how is the comparison done? Also, the authors do not provide an explanation on why the top 70 code types specifically were chosen for comparison.
- Page 12 – Line 12 onwards – It is not made clear how the neural spontaneous electrophysiological signals were applied to glioma cells, as there is not previous explanation on how these signals could be generated and applied to the cells.
- Page 12/13 – It is not very clear to me what the term “hijacking” constitutes experimentally, i.e., the authors claim that the neural networks are hijacked by glioma cells, however it is not made clear at what point in time does this effectively occur or what are the conditions to be achieved in order to consider that a certain network has been hijacked. Could that metric be a ratio in number of glioma cells versus neurons, or a specific number of physical connections between the 2 types of cells?
- Page 13 – Figure 5B – Why are the gamma and theta oscillation results presented as equal or very similar normal distributions to then present a percentage value? Can the authors report on an actual mean and std, as well as statistical analysis of this difference?

Minor:

- The resolution and quality of Figures is not appropriate. There are also many cases where the graphs presented have no information on axis, units and further details about the data and statistics. Examples: Figure 3i, Figure 4d, e, f, g, h, i
- Page 10, Line 20 – Typo “calcium hannel-”, instead of “calcium channel”.

Version 1:

Reviewer comments:

Reviewer #1

(Remarks to the Author)

In this revised manuscript, the authors do a good job to address all of the major concerns raised by this reviewer including presenting new data from experiments, and in particular, testing this model in another cell line U251. There are no further major revisions requested but a few minor revisions to improve clarity and organization of the overall manuscript.

- On line 257, the authors state 10,000 cells were seeded into the device, please clarify how many of these cells are tumor cells vs neurons

- It appears that the authors present for the first time their results regarding testing an additional cell line U251 and also checking tumor proliferation in the discussion section Lines 456-470. It may make more sense to first present this data in the results section as these are results and then discuss these findings or include them in the broader discussion that is present in the conclusion section.

- The authors discussion of the U251 cell line findings could be further improved for clarity. The authors state "recapitulating co-culture invasion intensity required re-extracting co-culture-derived 464 electrical signals". What exactly does this mean? Please clarify. Are the authors stating that a different U251-hijacked signal appeared to promote U251 invasiveness more so than the U87-hijacked signal?

- Can the authors clarify in the manuscript why it appears for all of the CMC graphs shown, all of the cell lines appear to converge at 3mm by day 7? Is 3mm the maximum value that can be measured by this assay? If so, it may be better to focus on which treatment arms or experimental conditions reach this maximum migration distance faster and more clearly explaining this distinction in the manuscript.

Reviewer #2

(Remarks to the Author)

The authors have addressed most of my comments. I have two quantity control questions/comments for the new figures:

- Are the groups statistically different in the new Figure S11?

- Quantification of the western blots in new Figure S8 is required

Reviewer #3

(Remarks to the Author)

I thank the authors for their thorough revisions and openness in addressing the various comments. I believe all major points have been address and that the article is now acceptable for publication. Congratulations on the great results and outcomes!

PS: I must just leave two very small comments and suggestions for the final version of the article, if possible:

- Please update the Materials & Methods section to provide details on the new studies performed on molecular signatures of cells (Figure S8);

- Please try to mention in either the M&M section or in the main text how cell viability was ensured during the tests. The authors provided a proper explanantion on their Responses document but I believe the article was not updated to include that.

COMMENTS TO AUTHOR:

Reviewer: #1

In this manuscript the authors develop a novel platform for measuring neural and glioma cell electrical signal patterns while utilizing machine learning algorithms to be able to reliably distinguish between neural-specific and glioma-specific bioelectrical signal patterns. The authors study glioma invasion in vitro with this platform and correlate tumor invasiveness to the specific electrical signal patterns being recorded by their platform. They find that specific neural networks can promote hyper-invasive behavior of brain tumor cells. They first applied neural-specific bioelectrical signals to glioma cells and found a modest increase in invasiveness although not the hyper-invasive phenotype. Next, they stimulated glioma cells with bioelectrical signal pattern of hijacked neural networks where glioma cells had modified the waveform characteristics of the bioelectrical signal patterns of neural cells. They found this glioma-altered neural network signal significantly accelerated tumor cell migration. Interestingly, the electrophysiological signal alone without any neurons was enough to increase tumor invasion.

This work is of significance to the field of glioblastoma pathophysiology and the interplay of neuron-glioma circuits. The authors do a good job of describing the current state of the literature in this field in their introduction and their work in this study appropriately expands upon our current knowledge. The platform utilized by the authors in this study is novel and provides key insights into how glioma cells alter the bioelectrical signal pattern of neurons and how these electrical circuits promote glioma invasion. The results and conclusions presented by the authors in this manuscript logically make sense within the context of the current state of the literature and our understanding of how neuron-glioma circuits function. The authors do a good job supporting their claims and conclusions with the experiments and figures provided in this manuscript. The methodology appears sound and definitely the correlation of the authors' in vitro experiments and immunofluorescence data to the bioelectrical signals recorded is convincing. A reviewer with a stronger background on computational biology should comment on the validity of the authors' machine learning algorithm calculations. This work meets the expected standards in this field.

In general, there is sufficient details provided in the methods for the work to be reproduced, please see suggestions below for some points for improvement. Overall, this is a well-written manuscript that is clear to understand. The author's present some very interesting results that are convincing based on the experiments completed and also make sense within the larger context of the established literature in this field.

We sincerely appreciate your time and the positive comments of our work. Our point-by-point responses are detailed below.

Suggestions:

- In the introduction, the authors could clearly state their hypothesis towards the end of this section. The authors talk about on lines 113-114 that mimicking electrophysiological signals can replicate tumor phenotype in vitro. However, they could state their hypothesis here which they actually talk about in their results and discussion section. Something along the lines of "Given our understanding that neuron-glioma interactions are bi-directional, we hypothesize that the electrophysiological signals produced by the interaction between these two cells types*

will have a greater effect on tumor invasiveness than the electrophysiological signals produced by neural cells alone” or something to that effect which seems to be essentially the central hypothesis of this paper. Presenting that in the introduction could help the reader better frame the results/discussion sections.

Our deepest gratitude for your careful reviewing and thoughtful suggestions. In the revised manuscript, we have added a state of the hypothesis at the end of the introduction to help the reader better frame the results/discussion sections. (page 6, lines 144-148)

- One experiment to potentially consider is testing different cell lines of glioma cells to determine if there are any differences between cell lines or if the same bioelectrical signaling patterns induce a similar degree of invasiveness across multiple cell lines.

We sincerely appreciate your suggestion. In the revised manuscript, we introduced a new human glioma cell line (U251) to investigate whether bioelectrical signaling patterns consistently promote tumor invasiveness across different cell lines. Our finding indicated that while U87-associated bioelectrical signals still moderately promoted tumor invasion, however, recapitulating the invasion intensity observed in co-culture systems necessitated re-extracting co-culture-derived electrical signals from U251, highlighting variability in bioelectrical signaling patterns among different cell lines (Fig. S11). Related discussion has been added on pages 18-19, lines 458-465.

Figure S11. Invasive behavior of human glioma U251 cells under four distinct bioelectrical conditions (Control, U87-hijacked signals, co-culture neural signals, or U251-hijacked signals).

- Additionally, it may not be too challenging to also look at glioma proliferation especially if all of the glioma cells are tagged with GFP. Measuring change in GFP intensity over time could be used to determine whether the tumors are growing faster as well as being more invasive as I suspect they may be.

We sincerely thank the reviewer for this insightful suggestion regarding the assessment of glioma proliferation dynamics using GFP-tagged cells. Following this valuable input, we performed longitudinal quantification of GFP intensity to monitor glioma cell proliferation dynamics, which further validated that tumors exhibit not only increased invasiveness but also accelerated growth rates (Fig S10). Relevant discussion has been added on page 18, lines 456-458.

Figure S10. Cell proliferation comparison between control and hijacked cells.

• Authors should provide the glioblastoma cell line or cell lines utilized in the co-culture experiments as it is not immediately clear which cell lines were used and whether these cell lines are derived from mouse, rat, or human.

We sincerely appreciate the reviewer's careful attention to this detail. In the revised manuscript, we have provided information on page 22, line 544. The glioblastoma cell line utilized in our experiments is U87 MG, a well-characterized human-derived glioblastoma cell line.

• For the experiment described in figure 4-5, the authors state glioma cells were exposed to electrical stimulation patterns but it is unclear how this electrical stimulation is applied to the tumor cells. More details on how the electrical stimulation was applied to the glioma cells could help clarify the methodology. Also, any information regarding duration of stimulation application would be helpful as well and provide further context for this experiment. For instance, was the stimulus provided for a total time of 30minutes or a couple of hours or did it vary?

Thanks for the comment. In the revised manuscript we have added more details on how electrical stimulation was applied to the glioma cells on page 27, lines 665-670.

The electrical stimulation was delivered via integrated microelectrodes within the microfluidic device (recording function of the stimulated microelectrodes was paused during stimulation). The stimulation duration was set to last throughout the entire experimental period, during which the stimulation was continuously applied to the tumor cells to simulate neuron-tumor interactions in co-culture.

Reviewer: #2

The authors developed an MEA + machine learning platform to study the interaction between GBM cells and neurons. This new platform can answer important questions in cancer neuroscience. The interesting findings from this novel system are that as long as the authors mimic delivering the neuronal electrical signal to the sole glioma culture, they observed a similar glioma migration-promoting effect as seen in the neuron-glioma co-culture, suggesting that neuronal activity-dependent factors may not be necessary in this phenomenon; however, the biological model system needs to be further developed and validated to support the conclusion.

We sincerely thank you for the time and careful consideration in reviewing our manuscript. We

have revised the manuscript accordingly. Our point-by-point responses are detailed below.

- It is unclear which GBM cell line the author used. Detailed information is not included in the methods. Regardless, using only one GBM cell line does not reflect the high degree of heterogeneity present in GBM. Therefore, multiple cell lines, particularly human cell lines, should be tested in this in vitro system.

Thanks for the helpful suggestions. In the revised manuscript, we have supplemented the cell source information in the methods section (page 22, lines 544). The glioblastoma cell line utilized in our experiments is U87 MG, a well-characterized human-derived glioblastoma cell line.

Furthermore, we incorporated an additional human glioma cell line (U251) into our experiments to validate the robustness of our primary findings. These data confirmed that neuron-glioma electrical signaling indeed enhances glioma migration (Fig. S11), consistent with our original observations. The updated results and corresponding analyses are presented in the revised manuscript at page 18, lines 458-465.

Figure S11. Invasive behavior of human glioma U251 cells under four distinct bioelectrical conditions (Control, U87-hijacked signals, co-culture neural signals, or U251-hijacked signals).

- It is unclear whether "normal state" and "hyper-invasive state" describe the same GBM cell line, just different behaviors in the MEA chambers. Additionally, what criteria are used to define these two states? - The authors claim that their system tests GBM cell invasion, which is debatable. The system forces glioma cells to enter the neuron-rich area. Therefore, GBM cell mechanical sensing and their attachment to the bottom both contribute to their migration toward the neurons. The authors should utilize molecular assays to distinguish between cells in the "normal state" and the "hyper-invasive state," and further evaluate whether the "hyper-invasive state" actually exhibits enriched molecular signatures in cell migration and invasion.

We appreciate your comments. We agree that further molecular analyses are needed to discriminate between these two functional states. In the revised manuscript, we have supplemented molecular characterization of the "normal state" and "hyper-invasive state," including detailed profiling of hyper-invasive behavior through differential expression in invasion-related proteins and RNAs. Molecular analyses (Fig. S8) revealed an invasion-associated transition characterized by downregulated E-cadherin, upregulated

Vimentin/Snail/Zeb1, and corresponding dysregulated EMT-related mRNAs (CDH1, VIM, SNAI1, ZEB1). These molecular hallmarks of invasiveness confirm the transition from a non-aggressive to a highly invasive state. Relevant discussion has been added on page 16, lines 402-407.

Figure S8. Invasion-related molecular signatures in hyper-invasive vs. control cells.

- Several sentences (lines 262-266, 310-312, 330-331) appeared as if they were novel findings by the authors; however, they had already been shown previously by multiple groups *in vivo* (Monje, Winkler; Deneen labs). The claims need to be softened, and prior studies need to be cited and discussed.

We sincerely appreciate your careful review of these specific sentences. In the revised manuscript, we have softened the claims by citing the prior studies from Monje, Winkler, and Deneen labs. (lines 328, 383, 409).

- In the methods, it's unclear how hippocampal neurons were isolated from E18 rat embryos. The system is somewhat unusual because GBM rarely forms in the hippocampus. What neuron subtypes are primarily present in this culture?

Thank you for your careful review of the methods section. We have revised the "Neuron Isolation and Culture" subsection (Page 22, Lines 551-557) to provide detailed protocols for hippocampal neuron isolation. Timed-pregnant E18 Sprague-Dawley rats were euthanized via CO₂ asphyxiation, embryos decapitated, and brains extracted into ice-cold HBSS (without Ca²⁺/Mg²⁺) with 10 mM HEPES (pH 7.4); hippocampi were microdissected under a stereomicroscope and dissociated via 15-minute 37° C incubation in papain, followed by gentle trituration; the cell suspension was centrifuged, resuspended in Neurobasal-A medium (supplemented with 2% B27, 0.5 mM L-glutamine, and 100 U/mL penicillin/streptomycin) for use.

Notably, even though GBM does not originate in the hippocampus, evidences have shown that hippocampal neurons do engage in functional interactions with GBM cells, and many relevant [1,2] studies typically involve implanting tumors into the hippocampus. Furthermore, hippocampal neural networks represent a uniquely well-characterized model in neurobiology: they encompass both glutamatergic (excitatory) and GABAergic (inhibitory) neuronal phenotypes, and their robust synaptic functionality and high experimental reproducibility have been consistently validated across decades of studies. We have expanded the Discussion section to explicitly note this context on page 22, lines 557-564. Thank you again for prompting this clarification.

References:

- [1] Venkatesh, H.S., Morishita, W., Geraghty, A.C. et al. Electrical and synaptic integration of glioma into neural circuits. *Nature* 573, 539 – 545 (2019).
- [2] Krishna, S., Choudhury, A., Keough, M.B. et al. Glioblastoma remodelling of human neural circuits decreases survival. *Nature* 617, 599–607 (2023).

Minor comments:

- In many places, there is no space before the first parenthesis of the reference.
- Lines 316-318 need citation.
- line 300, not sure why "pronounced" is used.
- Line 303, not sure what "hat glioma cells" means.
- Fig. 3d should specify which neuron marker is used in the image and/or legend

Thank you for your valuable comments. We have systematically checked the manuscript and added spaces before all reference parentheses to ensure formatting consistency.

Lines 316 – 318 citation: A relevant citation has been added to support this statement (line 390).

Line 300 "pronounced": We have changed “pronounced” to “less striking”. So the text will be "However, this observed increase in invasion was less striking than..." .

Line 303 "hat glioma cells": This was a typo and has been corrected to "that glioma cells".

Fig. 3d neuron marker: The legend for Fig. 3d has been updated to specify that the neuron marker (NeuN) used.

We appreciate your attention to these details, which have helped improve the clarity and rigor of our manuscript.

Reviewer #3

The manuscript deals with a proposed machine learning-based electrophysiological platform that combines custom microfluidics with real-time decoding of neural-tumor signal interactions. The manuscript claims that glioma cells do not passively respond to neural activity but instead selectively hijack specific neural electrical patterns. The authors claim that this selective entrainment leads to synchronized firing between neurons and tumor cells, which in turn drives hyper-invasive tumor behavior. Furthermore, the authors demonstrate that direct stimulation of glioma cells with these hijacked patterns is sufficient to induce aggressive invasion. The manuscript presents this platform as a novel tool for dissecting dynamic tumor-neural communication and suggests its potential for translational applications targeting bioelectrical signaling in glioma progression.

The manuscript is well written and presents a wide variety of experiments and data analytics endeavors. However, it does not typically provide full contextualization and details about the experiments being performed. The developed platform is quite complete, and it is shown that it is capable to precisely acquire signals from glioma and neurons. Despite this, I believe there are some experiments missing regarding the assessment of cell viability, cell-cell interaction and even the claimed effect on tumor cells, as, for example, it was not show if glioma cells would move across the device even if neurons were not present. Moreover, certain concepts like what constitutes “hijacking” or why it certain waveforms might be more relevant than others are not properly explained. Because of this, I believe that further experiments and results must be added in order to reach publishable status, as well as editing on the text to make many of

the results presented more clearly understandable. Please, find below a list of comments to be addressed in the next version of the manuscript.

We sincerely appreciate your time and the valuable comments of our work. We have revised the manuscript based on your suggestions, and our point-by-point responses to your comments are detailed in the following sections.

Comments:

- Page 3, Line 7-11: During Introduction, more context should be provided on the relationship between cancer cell exploitations of ion channels and their “high proliferative rate, migratory behavior, and invasiveness(14). ” . Without further contextualization, it does not become clear why, for example, mutations in ion channels of 90% of glioblastoma should be related to aggressiveness.

We thank the reviewer for this valuable comment. We have expanded the Introduction to clarify how ion channel exploitation drives cancer hallmarks on page 3, lines 74-88.

Mechanistically, ion channels regulate proliferation via membrane potential/ionic homeostasis control of cell cycle checkpoints (e.g., voltage-gated sodium channels maintain glioblastoma stemness in G0 phase) [1,2], while calcium-activated potassium channel (BK) correlate with glioma grade and directly regulate growth [3]. For migration/invasion, coordinated K⁺/Cl⁻ efflux via BK channels enables volume shrinkage critical for navigating brain tissue [4,5], with peripheral glioblastoma cells showing calcium transient-synchronized collective invasion [6]. Notably, glioblastomas exhibit pathological ion channel reprogramming (e.g., near-complete Kir4.1 loss and BK upregulation) [7], linking 90% ion channel mutations to aggressive phenotypes through disrupted differentiation and enhanced plasticity [8].

References:

1. Prevarskaya, N., R. Skryma, and Y. Shuba, Ion Channels in Cancer: Are Cancer Hallmarks Oncochannelopathies? *Physiol Rev*, 2018. 98(2): p. 559-621.
2. Giamello, F., et al., Modulating voltage-gated sodium channels to enhance differentiation and sensitize glioblastoma cells to chemotherapy. *Cell Commun Signal*, 2024. 22(1): p. 434.
3. Michelucci, A., et al., Hypoxia, Ion Channels and Glioblastoma Malignancy. *Biomolecules*, 2023. 13(12).
4. Soroceanu, L., T.J. Manning, Jr., and H. Sontheimer, Modulation of glioma cell migration and invasion using Cl⁻ and K⁺ ion channel blockers. *J Neurosci*, 1999. 19(14): p. 5942-54.
5. McFerrin, M.B. and H. Sontheimer, A role for ion channels in glioma cell invasion. *Neuron Glia Biol*, 2006. 2(1): p. 39-49.
6. Xu, M., et al., A temporal examination of calcium signaling in cancer- from tumorigenesis, to immune evasion, and metastasis. *Cell Biosci*, 2018. 8: p. 25.
7. Liu, J., et al., Potassium channels and their role in glioma: A mini review. *Mol Membr Biol*, 2019. 35(1): p. 76-85.
8. Hanahan, D., Hallmarks of Cancer: New Dimensions. *Cancer Discov*, 2022. 12(1): p. 31-46.

- Page 3, Line 21-22: When stating that “excitatory postsynaptic currents (EPSCs), (lead) to glioma cell membrane depolarization and promoting tumor proliferation”, it would help readers if more details about glioma as a disease, its onset, how does it typically expand and what are some of its key proliferative mechanisms, for example.

Thank you for your valuable suggestion. We have expanded the description of glioma as a disease and its key proliferative mechanisms in the revised manuscript (Page 4, Lines 101-112). Gliomas are primary malignant CNS tumors with high invasiveness, recurrence, and poor prognosis. Onset involves genetic/epigenetic dysregulation, and key proliferative mechanisms include dysregulated signaling cascades (e.g., PI3K/Akt/mTOR, MAPK/ERK), ion channel dysfunction, and tumor microenvironment (TME) remodeling. Critically, Venkataramani et al. [1] identified functional glutamatergic synapses ("neurogliomal synapses") between neurons and glioma cells: presynaptic glutamate release activates postsynaptic AMPA receptors on glioma cells, triggering excitatory postsynaptic currents (EPSCs), membrane depolarization, and calcium influx [2]. This neuronal-glioma signaling directly promotes GSC proliferation and invasion, a mechanism validated by preclinical studies showing AMPA receptor antagonists inhibit tumor growth.

References

1. Venkataramani, V., Tanev, D.I., Strahle, C. et al. Glutamatergic synaptic input to glioma cells drives brain tumour progression. *Nature* 573, 532 – 538 (2019).
2. Venkatesh, H.S., Morishita, W., Geraghty, A.C. et al. Electrical and synaptic integration of glioma into neural circuits. *Nature* 573, 539 – 545 (2019).

- Page 5, Line 22-25 (and Method section): Can the authors provide further information regarding the culturing conditions, specifically, viability assessment along experimental time and typical numbers of cultured cells? Also, did the authors assess the level of adhesion of cells to the coated electrodes, i.e., was it assessed if all cells or only a small fraction of the seeded cells stayed within the device?

We sincerely appreciate your constructive comments. Further information has been provided in the revised manuscript. Specifically, the typical number of cultured cells was 10,000 and viability assessment has been supplemented in Fig. S4 (which showed that the cell survival rate remained above 80% after 7 days of culture). Our observations also confirmed that almost all seeded cells were retained within the device. Related discussion has been provided on page 10, lines 257-258.

Figure S4. Cell viability after 7 days of culture. Green: Active cells expressing GFP; Blue: DAPI staining. Scale bar: 200 μ m.

- Page 7 - Can the authors provide more information regarding how measurements were performed and how the electrode array was designed? Also, further details on calibration

experiments or efforts to understand how the signals might vary under different conditions would be valuable. Moreover, on the filtering criteria provided, how is the factor applied determined?

We thank the reviewer for these questions. The measurement was performed in the customized device fitted with an array of electrodes on microchannels. The electrode array (Fig. 2b) was designed in AutoCAD 2020 (Autodesk, Inc.) and fabricated via a standard photolithography-based lift-off process. Au electrodes (100 μm width \times 3 mm length, 200 μm center-to-center spacing) were patterned by depositing a 100 nm-thick gold layer via magnetron sputtering (ISC-150, SuPro Instruments), with the reference electrode placed at the distal end of the channel, 4.6 mm from the last active electrode. Electrophysiological signals were amplified using a customized capture system with a 1000x gain and sampled at 30 kHz to capture both slow local field potentials and rapid spiking neural activity.

Calibration experiments included electrode resistance calibration, noise characterization, and inter-device consistency calibration. For each recording channel, baseline noise was measured in a cell-free "blank" culture chamber (identical to experimental chambers but without cells) over a 24-hour period. To eliminate device-specific artifacts, three replicate MEA systems (Devices A, B, and C) were cross-validated using a standardized electrical signal source (sinusoidal wave: 100 μV amplitude, 10–500 Hz frequency range). For each device, two key metrics were assessed: signal amplitude deviation (<2% variation across devices after gain correction) and temporal synchronization, where time delays between input and recorded signals were adjusted via FPGA-based timestamp alignment (resulting in residual delays <1 ms). Devices failing to meet these criteria (e.g., >5% amplitude deviation) underwent hardware recalibration (e.g., amplifier gain adjustment) before experimental use. The filtering factor (3–5 \times SD) is primarily grounded in classical neuroscience experience, with supplementary validation from the statistical properties of signal noise. All related information has been provided on pages 20-21, lines 493-515.

- Page 8 - Can the authors report on control experiments where the neurons were not initially present on the chamber before glioma cells are added? It would be critical to confirm if the presence of the neurons within the chamber has an effect on the movement observed.

We thank the reviewer for the question. In the revised manuscript, we have further discussed the control experiment data for the scenario where neurons were not initially present in the chamber. The results indicate that glioma cells exhibit a baseline invasive behavior in the absence of neurons (defined as "Normal state", Fig. 3i). Notably, after the addition of neurons, the invasion rate of tumor cells was observed to increase significantly (defined as "Hyper-Invasive", Fig. 3i). This control experiment validates the effect of neuron presence on tumor movement.

Figure 3i. Migration behavior (cell migration center) of glioma cells in neuron-free (Normal state) vs. neuron-seeded (Hyper-Invasive state).

- Page 8 - It is not clear to me what constitutes the “hyper-invasive behavior”. How do the authors identify this behavior? What metrics, signals or other, are used to classify them as such?

Thank you for your question. To define "hyper-invasive behavior," we first established baseline collective invasion velocity of tumor cells under microenvironmental conditions devoid of extrinsic stimuli, quantified by tracking cell migration dynamics over time. This baseline was operationally defined as "normal invasive behavior." Upon exposure to pro-invasive cues (e.g., neuron-derived signaling or electrical stimulation), we observed a statistically significant increase in collective invasion velocity compared to baseline; this phenotype was designated "hyper-invasive behavior." Furthermore, hyper-invasive behavior was characterized by molecular signatures (page 16, lines 402-407), including enhanced epithelial-mesenchymal transition (EMT) (downregulated E-cadherin, upregulated Vimentin/Snail/Zeb1), and dysregulated expression of EMT-related mRNAs (CDH1, VIM, SNAI1, ZEB1, Fig. S8). These molecular signatures have been well-documented to correlate with tumor invasiveness, supporting the phenotypic shift from a quiescent to an aggressively invasive state.

Figure S8. Invasion-related molecular signatures in hyper-invasive vs. control cells.

- Page 8 - Can the authors comment on the ratio of glioma cells that started movement versus the population that stayed within the input sample? Figure 3 does not make it clear if all the cells or only a small fraction actually initiated movement across the device.

Thank you for your question. Not all glioma cells traverse the device, while only a subset completes migration and the majority persists near the primary tumor site or along the migration

path. However, precise quantification of the motile-to-retained cell ratio in input samples remains challenging due to continuous tumor cell proliferation and migration causing dynamic temporal fluctuations in total and migratory counts, which are further complicated by delayed invasion (movement initiating at later rather than immediate time points).

- Page 8/9 - *Can the authors comment on the observed movement of glioma cells, i.e., if their movement followed any specific trend, sequence, morphological change or any other behavior that could also be important to highlight?*

Thank you for your question. Observations showed that glioma cells primarily exhibited faster invasion in hyper-state, however, their migratory behavior did not exhibit distinct trends, sequential patterns. Although no significant changes in cell morphology or invasion patterns were detected, molecular analyses (Fig. S8) demonstrated an invasion-associated transition—characterized by downregulated E-cadherin and upregulated Vimentin, Snail, and Zeb1—accompanied by dysregulated expression of EMT-related mRNAs (CDH1, VIM, SNAI1, ZEB1). These molecular signatures have been well-documented to correlate with tumor invasiveness, supporting the phenotypic shift from a quiescent to an aggressively invasive state.

- Page 8/9 - *During the 7 day experiment, how was viability of cells ensured if the liquid within the chamber was stopped in order to ensure no further movement induced into the channel?*

Thank you for your comment. To maintain cell viability during the 7-day experiment, we periodically replaced half of the culture medium within the microfluidic chamber every 6–12 hours, which effectively ensured cell survival despite the cessation of liquid flow to prevent additional movement in the channel. As shown in Fig. S4, viability assessment showed that the cell survival rate remained above 80% after 7 days of culture. Related discussion has been provided on page 10, lines 258-259.

Figure S4. Cell viability after 7 days of culture. Green: Active cells expressing GFP; Blue: DAPI staining. Scale bar: 200 μm .

- Page 9 - *It is confusing to use “i” as a variable for both the electrode serial number and the center-to-center distance of electrodes. Also, it is not clear the difference between “i” and “d” on that regard? Moreover, the authors should provide, even if in supplementary and for a small set of experiments, a comparison between the classical approach highlighted of individual cell tracking and their proposed approach, in order to showcase the differences and potential advantages of the proposed method.*

Thank you for your question. We apologize for any confusion caused by previous description, and we have provided more detailed information in the revised manuscript (page 10, lines 268-272). Specifically, the variable "i" is exclusively defined as the electrode serial number, while

"d" denotes the center-to-center distance between adjacent electrodes (200 μm in our device). Based on the reviewers' comments, we have also added a comparison between the classical single-cell tracking approach and our proposed population-level analysis. Compared to traditional single-cell tracking, this population-level analysis largely reduces data variability due to high intercellular differences in migration speed within the tumor population (**Fig. S5**).

Figure S5. Comparison of Standard Error between population and single-cell tracking.

- Page 10 - Figure 4B - It is not very clear what is a neuron versus glioma cell in the images provided. For example, it seems that the Calcium trigger goes from Neuron (1) to Glioma (2) around the Glioma (3), while at the same time there are other cells (?)/locations with high signal as well that are not annotated or receive any comments. I would ask the authors to better annotate the Figure, as well as providing a better discussion of the phenomena being observed in it. - Page 11, Line 2 - Can the authors comment why the synchronized firing might have went from Neuron (1) to Glioma (2) to (3), instead of (1) to (3), as Glioma (3) is much closer to Neuron (1)? Also, why does Glioma (3) have less firing events compared with the other cells? And can the authors analyze and provide the data measured at other cells that were found surrounding these 3 cells? Is this firing effect short and local, or does it propagate further into other cells?

Thank you for your question. Following the reviewer's suggestion, we have enhanced the figure annotations and expanded a detailed discussion on page 13, line 316-326 in the revised manuscript. Please note that Cell 2 and Cell 3 in the original figure have been renumbered as Cell 3 and Cell 4 in the revised version, and subsequent discussions will refer to the updated numbering.

As shown in Fig. 4b, Cell 1 initially exhibits firing, which propagates to Cell 2, then subsequently to Cells 3 and 5. Upon simultaneous stimulation from Cells 1 and 3, Cell 4—which appears to have a higher activation threshold—is activated and engages in co-firing with Cells 1 and 3. It should be noted that neural network conduction is not solely determined by spatial distance but also depends on signal intensity and inherent cellular electrophysiological thresholds. It can be observed that Cell 4 itself exhibits reduced firing events, suggesting it may have a higher activation threshold and consequently lower excitability. Consequently, despite Cell 4's closer proximity to Cell 1, the signal preferentially activates Cell 3, and only the convergent stimulation from Cell 1 and Cell 3 does Cell 4 become activated and functionally incorporated into the form co-firing network. Also, as suggested by the reviewer, we provided electrical signals from cell 2 and cell 5 (Fig.S7), no obvious co-firing was identified with Cell1, indicating these firing effects are locally transient with no further propagation.

Figure 4b. Cellular calcium signal transduction diagram

Figure S7. Signal display of Cells 1, 2, and 5.

- Page 11 - Line 22 - The authors must properly explain how the “neural codes” were identified and distinguished into more than 1000 types. It is not clear what metrics were used to compare them or how they were evaluated. Also, how was comparison across different experiments and observed glioma behaviors for ex?

We appreciate the valuable suggestions. In the revised manuscript, we have updated the explanation of how the “neural codes” were identified and distinguished on pages 26, lines 648-658.

Neural codes were identified from electrophysiological activities exceeding baseline levels during functional behaviors of cells, with similar waveforms grouped into fixed and unique neural code types to enable consistent tracking across experiments and comparison of temporal usage patterns under varying conditions. Waveforms were classified as follows: Common-mode noise was firstly eliminated by subtracting overall signal fluctuations from each recording site. Next, bidirectional application of zero-phase Butterworth filtering (1~3000Hz) was applied to mitigate phase distortion. The active signals were then extracted and peak-aligned to time 0, calibrating their peak amplitudes to the 0-time reference point, before proceeding to waveform similarity analysis. Waveform similarity comparison and classification were performed by digit-by-digit comparison from the 0-time point outward, with label consistency evaluated against a predefined threshold (10%). Waveforms with $\geq 90\%$ label consistency were grouped into a fixed and unique neural code type. Each new waveform was cross-checked against existing template classes in the neural code library, and unmatched waveforms were designated as new neural code types. This integrated methodology enabled the identification of over 10000 neural code types and facilitated reliable cross-experimental comparisons.

- Page 12 - Figure 4i - I am unfamiliar with the type of plot utilized to compare the simulated signal with the nerve stimulation signal. Can the authors comment on what is being evaluated and how is the comparison done? Also, the authors do not provide an explanation on why the top 70 code types specifically were chosen for comparison.

Thank you for your valuable comment. Figure 4i presents the relative occurrence frequency of code types with the most significant changes and visualizes their proportional distribution via a pie chart, enabling clear comparison of proportional structure changes across conditions. The top 70 code types were selected as they collectively account for most of the total variation, balancing comprehensiveness (capturing major changes) and figure readability, thus adequately representing overall trends of interest. Related discussions have been provided in the revised manuscript on page 14, lines 364-365.

- Page 12 - Line 12 onwards - It is not made clear how the neural spontaneous electrophysiological signals were applied to glioma cells, as there is not previous explanation on how these signals could be generated and applied to the cells.

Thank you for the comment. Neural spontaneous electrophysiological signals were collected from a dedicated device containing pure neural networks (without tumor cells), and required signals were then applied to a separate device with pure tumor cells via the microelectrodes at the bottom of the microfluidic chamber. Thus, we can evaluate the effect of pure neural spontaneous electrophysiological signals on tumor behavior. Related information has been added on page 26, line 661-670.

- Page 12/13 - It is not very clear to me what the term "hijacking" constitutes experimentally, i.e., the authors claim that the neural networks are hijacked by glioma cells, however it is not made clear at what point in time does this effectively occur or what are the conditions to be achieved in order to consider that a certain network has been hijacked. Could that metric be a ratio in number of glioma cells versus neurons, or a specific number of physical connections between the 2 types of cells?

We thank the reviewer for your professional suggestions. In the revised manuscript, we have systematically evaluated glioma cell behavior across varying glioma-to-neuron ratios to determine a metric ratio in number of glioma cells versus neurons. As shown in **Fig. S12**, the enhanced hyper activity of the tumor increased with the increase in the proportion of glioma cells to neurons. When the ratio was 1:1, the enhanced hyper activity of the tumor reached the highest level. Then, as the proportion of neurons further increased, this "hijacking" state gradually disappeared. Related discussions have been provided in the revised manuscript on page 19, lines 465-470.

Figure S12. Glioma cell behavior across varying glioma-to-neuron ratios.

- Page 13 - Figure 5B - Why are the gamma and theta oscillation results presented as equal or very similar normal distributions to then present a percentage value? Can the authors report on an actual mean and std, as well as statistical analysis of this difference?

We are grateful for your valuable comment. In the revised manuscript, we have presented the data as an actual mean and std, as well as statistical analysis of this difference (page 39, Fig. 5).

Figure 5B. Hijacking by hyper-invasive gliomas results in enhanced activity within the gamma (30-100 Hz) and theta (4-8 Hz) frequency bands of neural signals.

Minor:

- The resolution and quality of Figures is not appropriate. There are also many cases where the graphs presented have no information on axis, units and further details about the data and statistics. Examples: Figure 3i, Figure 4d, e, f, g, h, l

- Page 10, Line 20 - Typo "calcium hannel-" , instead of "calcium channel" .

We appreciate your attention to these details. All figures (including Figure 3i, 4d, 4e, 4f, 4g, 4h, and 4l) have been revised. Axes, units, data details, and statistical information have been added to ensure clarity and reproducibility (Page 38).

Due to Microsoft Word constraints, image resolution is compromised. We will provide original high-resolution images for proper publication.

The typo "calcium hannel-" on Page 10, Line 20 has been corrected to "calcium channel" in the revised manuscript.

COMMENTS TO AUTHOR:

Reviewer: #1

In this revised manuscript, the authors do a good job to address all of the major concerns raised by this reviewer including presenting new data from experiments, and in particular, testing this model in another cell line U251. There are no further major revisions requested but a few minor revisions to improve clarity and organization of the overall manuscript.

- On line 257, the authors state 10,000 cells were seeded into the device, please clarify how many of these cells are tumor cells vs neurons.

We sincerely appreciate your constructive comments. Further information has been provided in the revised manuscript. The ratio of tumor cells to nerve cells is generally 1:1. (lines 220-221)

- It appears that the authors present for the first time their results regarding testing an additional cell line U251 and also checking tumor proliferation in the discussion section Lines 456-470. It may make more sense to first present this data in the results section as these are results and then discuss these findings or include them in the broader discussion that is present in the conclusion section.

Thank you for your suggestion. In the revised manuscript, the data regarding the U251 cell line has been moved to the Results section, and the related findings have been further discussed in the Discussion section (lines 422-431).

- The authors discussion of the U251 cell line findings could be further improved for clarity. The authors state “recapitulating co-culture invasion intensity required re-extracting co-culture-derived 464 electrical signals”. What exactly does this mean? Please clarify. Are the authors stating that a different U251-hijacked signal appeared to promote U251 invasiveness more so than the U87-hijacked signal?

We appreciate your constructive feedback. To clarify the findings regarding the U251 cell line: The electrical signals extracted from U87 cells can promote U251 invasion; however, the signals derived from U251 cells themselves exhibit a stronger promotional effect on U251 invasiveness. The revised description has been updated in the corresponding pages of the revised manuscript (lines 428-431).

- Can the authors clarify in the manuscript why it appears for all of the CMC graphs shown, all of the cell lines appear to converge at 3mm by day 7? Is 3mm the maximum value that can be measured by this assay? If so, it may be better to focus on which treatment arms or experimental conditions reach this maximum migration distance faster and more clearly explaining this distinction in the manuscript.

Thank you for raising this important point. The 3 mm distance in the CMC graphs represents the maximum collective migration distance of the cells used in this project. Our analysis of migration speed is actually based on the order in which each group reaches this maximum distance or the migration distance achieved within the same time period, which is fully consistent with your suggestion.

Reviewer #2

The authors have addressed most of my comments. I have two quantity control questions/comments for the new figures:

- Are the groups statistically different in the new Figure S11?

Thank you for your question. There is no significant difference between the U251 hijacked group and the Co-culture group. However, these two groups show differences from the other groups in the first 5 days, but no differences are observed on days 6-7. A relevant discussion has been added in the revised manuscript (lines 425-428).

- Quantification of the western blots in new Figure S8 is required.

We appreciate the reviewer's valuable comment. We have now provided the densitometric quantification of the western blots in new Figure S8, based on three independent biological replicates.

Reviewer #3

I thank the authors for their thorough revisions and openness in addressing the various comments. I believe all major points have been addressed and that the article is now acceptable for publication. Congratulations on the great results and outcomes!

PS: I must just leave two very small comments and suggestions for the final version of the article, if possible:

- Please update the Materials & Methods section to provide details on the new studies performed on molecular signatures of cells (Figure S8);

Thanks for your comment. This additional information has been incorporated on lines 734-756 of the revised manuscript.

- Please try to mention in either the M&M section or in the main text how cell viability was ensured during the tests. The authors provided a proper explanation on their Responses document but I believe the article was not updated to include that.

We sincerely appreciate your constructive comments. We have supplemented the revised manuscript with related details in lines 546-549. Specifically, the culture medium within the microfluidic chip was regularly replaced to sustain cell activity, and cell viability was monitored by periodic observation of cellular morphology and electrophysiological signals to ensure normal cellular function.